# A fluorogenic cyclic peptide for imaging and quantification of drug-induced apoptosis

Nicole D. Barth[1,7], Ramon Subiros-Funosas[1,7], Lorena Mendive-Tapia[1], Rodger Duffin[1], Mario A. Shields [2], Jennifer A. Cartwright [1], Sónia Troeira Henriques[3,6], Jesus Sot[4], Felix M. Goñi[4], Rodolfo Lavilla [5], John A. Marwick[1], Sonja Vermeren [1], Adriano G. Rossi[1], Mikala Egeblad [2], Ian Dransfield[1✉] & Marc Vendrell [1✉]

Programmed cell death or apoptosis is a central biological process that is dysregulated in many diseases, including inflammatory conditions and cancer. The detection and quantification of apoptotic cells in vivo is hampered by the need for fixatives or washing steps for non-fluorogenic reagents, and by the low levels of free calcium in diseased tissues that restrict the use of annexins. In this manuscript, we report the rational design of a highly stable fluorogenic peptide (termed **Apo-15**) that selectively stains apoptotic cells in vitro and in vivo in a calcium-independent manner and under wash-free conditions. Furthermore, using a combination of chemical and biophysical methods, we identify phosphatidylserine as a molecular target of **Apo-15**. We demonstrate that **Apo-15** can be used for the quantification and imaging of drug-induced apoptosis in preclinical mouse models, thus creating opportunities for assessing the in vivo efficacy of anti-inflammatory and anti-cancer therapeutics.

[1] Centre for Inflammation Research, University of Edinburgh, EH16 4TJ Edinburgh, UK. [2] Cold Spring Harbor Laboratory, Cold Spring Harbor, NY 11724, USA. [3] Institute for Molecular Bioscience, The University of Queensland, Brisbane, Queensland 4072, Australia. [4] Instituto Biofisika (CSIC, UPV/EHU) and Departamento de Bioquímica, Universidad del País Vasco, Campus de Leioa, 48940 Leioa, Spain. [5] Laboratory of Medicinal Chemistry and Institute of Biomedicine U. Barcelona (IBUB), Faculty of Pharmacy, University of Barcelona, 08028 Barcelona, Spain. [6]Present address: School of Biomedical Sciences, Queensland University of Technology, Translational Research Institute, Brisbane, QLD 4102, Australia. [7]These authors contributed equally: Nicole D. Barth, Ramon Subiros-Funosas. ✉email: i.dransfield@ed.ac.uk; marc.vendrell@ed.ac.uk

Programmed cell death (apoptosis) is pivotal for maintenance of tissues and the regulation of inflammatory diseases. In contrast to necrosis, plasma membrane integrity is preserved during apoptosis, preventing the release of intracellular contents that can damage tissue and trigger inflammatory responses. Tissue phagocytes recognize apoptotic cells, providing a mechanism for the safe disposal of apoptotic material. Critically, excessive apoptosis or failure to clear apoptotic material results in secondary necrosis with the release of pro-inflammatory intracellular contents. Therefore, the presence of apoptotic cells represents a biomarker of the extent of tissue injury and correlates to the progression, resolution, and treatment of inflammatory conditions and cancer[1,2].

A sequence of morphological and biochemical changes occurs during apoptosis (e.g., phospholipid exposure, caspase activation, mitochondrial dysfunction, DNA fragmentation)[3–6]. Optical reagents for the detection of these events have been reported, but many probes alter cellular behavior, limiting their use for non-invasive detection of apoptosis in vivo. In particular, these limitations impede studies under physiological conditions (e.g., intravital imaging) and in situ assessment of therapy-induced apoptosis in preclinical models. At early stages of apoptosis, the plasma membrane undergoes profound remodeling and the activation of the scramblase Xkr8 promotes external exposure of phospholipids containing phosphatidylserine (PS)-headgroups and other phospholipids, which are normally restricted to the intracellular leaflet[7]. Reagents that bind to PS exposed on the plasma membrane (e.g., annexins)[4,8] have advantages over those monitoring intracellular changes (e.g., caspase activation, DNA fragmentation) because they do not require cell permeabilization or fixation. However, fluorescently-labeled annexins and polarity-sensitive annexins (pSIVA) require high concentrations of free $Ca^{2+}$ (>1 mM) to permit optimal phospholipid binding[4,5], limiting their performance in the hypocalcemic regions that are common in diseased tissues. Furthermore, annexins inhibit the engulfment of apoptotic cells by phagocytes[9], which precludes quantitative analysis of therapy-induced apoptosis in vivo.

In this study, we describe the rational design, optimization and validation of a fluorogenic peptide (termed **Apo-15**) to bind negatively-charged phospholipids exposed on apoptotic cells. **Apo-15** behaves as a universal apoptosis probe in that it detects apoptotic cells from multiple origins and in a broad range of experimental conditions. Furthermore, we demonstrate that **Apo-15** enables fluorescence imaging of apoptosis in vivo and quantification of drug-induced apoptosis in two different preclinical mouse models of acute lung injury (ALI) and breast cancer.

## Results

**Rational design and synthesis of fluorogenic peptides.** We designed cyclic amphipathic peptides (termed apopeptides) which we predicted would bind to phospholipids translocated to the outer leaflet of apoptotic cell membranes. We focused on small cyclic peptides because they offer several advantages: (1) resistance to proteolytic cleavage and oxidative conditions for in vivo studies, (2) tunability of the properties by changing the amino acid sequence, which would allow us to optimize binding to apoptotic cells, (3) compatibility with fluorogenic amino acids for wash-free imaging, and (4) smaller size than proteins for improved tissue accessibility. Apopeptides were designed to contain combinations of polar and hydrophobic amino acids to identify sequences that would bind to apoptotic cell membranes and remain unable to bind viable cells. All the apopeptides incorporated the environmentally-sensitive Trp-BODIPY fluorophore[10–12], which emits bright fluorescence after binding to provide optimal discrimination between viable and apoptotic cells (Fig. 1a, c).

A total of 15 apopeptides were prepared using solid-phase and solution peptide synthesis (Fig. 1b). In addition to Trp-BODIPY, apopeptides included tryptophan (W), phenylalanine (F), leucine (L), valine (V), and isoleucine (I) as hydrophobic residues, and lysine (K), glutamic acid (E), and arginine (R) as polar residues. In all cases, one glycine was included at the C-terminal end to facilitate head-to-tail cyclization (for enhanced resistance to proteolysis[13,14]), and to avoid stereoisomeric mixtures. Unprotected amino acids were used for the hydrophobic residues, whereas protected amino acids [e.g., Fmoc-Lys(Z)-OH, Fmoc-Glu(Bzl)-OH, Fmoc-Arg(NO₂)-OH] were used for the polar residues. The solid-phase peptide elongation was performed on 2-chlorotritylchloride polystyrene resin using conventional protocols and mild acidic cleavage conditions [i.e., TFA:DCM (1:99)][15]. Peptides were cyclized in a head-to-tail fashion using COMU as the condensation reagent[16], and side-chain protecting groups were removed to afford all apopeptides in high purities (>97%) after HPLC purification (Supplementary Table 1).

The sequences and physicochemical properties of all apopeptides are summarized in Fig. 1b. We sought to identify those sequences that would rapidly binding to apoptotic cells with minimal labeling of viable cells. cLac-BODIPY, a cyclic peptide able to bind apoptotic bodies but not apoptotic cells[11], was included for comparison. Using a flow cytometry-based assay, we examined the time-dependent emission of apopeptides in mixtures of both apoptotic and viable cells. We used human neutrophils cultured in vitro for 18 h, in which a large fraction of the cells (≥50%) undergo tissue culture-induced apoptosis[17]. Apoptotic and viable populations were defined by positive and negative staining with AF647-Annexin V in media containing 2 mM $CaCl_2$, respectively. First, we assessed how polar residues influenced binding by comparing two peptides with similar molecular weight and clog $P$ (Supplementary Table 1 and Fig. 1b), but with either negatively-charged (**Apo-0**) or positively-charged (**Apo-2**) residues. We chose glutamic acid (E) as a negatively-charged amino acid over aspartic acid to avoid synthetic complications due to the potential formation of aspartimides[18]. **Apo-2** showed selective binding to apoptotic cells over viable cells when compared with **Apo-0**, indicating the importance of positive charges for binding to negatively-charged phospholipids on apoptotic cell membranes. Next, we generated amphipathic peptides containing positively-charged amino acids and other residues that would alter binding to apoptotic cell membranes[19,20]. Specifically, we synthesized apopeptides to examine the influence of (1) aromatic vs non-aromatic hydrophobic residues (**Apo-3**, **4**, and **Apo 9–10**), (2) alternate vs sequential charges (**Apo 5–8**), and (3) overall polarity as determined by clog $P$ values (**Apo 11–14**).

Temporal analysis indicated that recognition of apoptotic cells occurred rapidly, with most apopeptides showing ≥80% of full binding in <4 min (Supplementary Table 2). From the screening, we quantified parameters that defined the selectivity and affinity of apopeptides: (1) preferential binding to apoptotic vs viable cells as fluorescence fold increase ($Ff$), (2) background fluorescence on viable cells ($Bf$), and (3) retention of binding upon washing ($Rt$) (Fig. 1b). Several apopeptides showed good discrimination between apoptotic and viable cells ($Ff ≥ 10$: **Apo-2**, **3**, **6**, and **8**), negligible fluorescence on viable cells ($Bf ≤ 3$: **Apo-0**, **4**, **11**, **12**, **13**, and **14**) and reasonable resistance to washing ($Rt ≥ 40\%$: **Apo-2**, **3**, **6**, **8**, and **11**). Overall, peptide polarity or clog P was related to retention of labeling (Fig. 1b). Although markedly polar peptides (clog $P < −4$, **Apo-13** and **Apo-14**) bound with fast kinetics, their signal was lost after washing. In contrast, less polar peptides (clog $P > −1$, **Apo-11** and **Apo-12**) displayed slower binding rates but their binding was relatively resistant to washing. These analyses suggested that apopeptides with balanced polarity (clog $P$ between −1 and −4) exhibited better labeling. **Apo-8** presented the highest retention of signal but also showed the highest binding to viable cells. Our

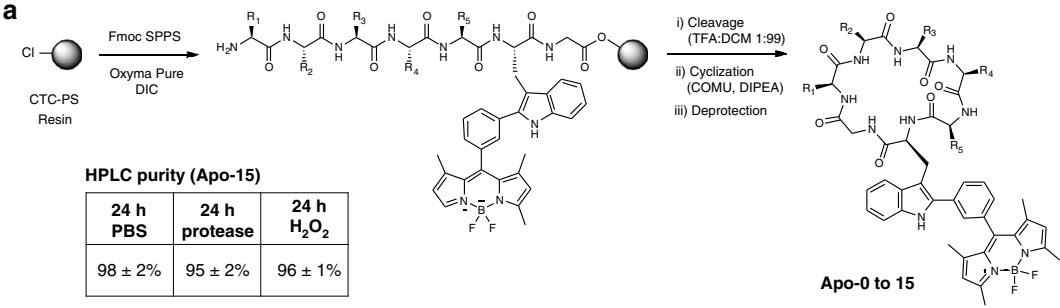

| HPLC purity (Apo-15) | | |
|---|---|---|
| 24 h PBS | 24 h protease | 24 h H₂O₂ |
| 98 ± 2% | 95 ± 2% | 96 ± 1% |

| Peptides | R₁–R₅ | clog P | Fold increase (Ff) [apoptotic vs viable] | Background (Bf) [viable cells] | Retention (Rt, %) |
|---|---|---|---|---|---|
| Apo-0 | EEEWW | –2.97 | <1 | 2.4 | n.a. |
| Apo-2 | KKKWW | –2.78 | 18 | 6.6 | 62 |
| Apo-3 | KKKWF | –2.93 | 14 | 4.7 | 47 |
| Apo-4 | KKKFF | –3.08 | 6 | 3.0 | 15 |
| Apo-5 | KWKWK | –2.78 | 7 | 3.6 | 8 |
| Apo-6 | KKWWK | –2.78 | 18 | 5.6 | 40 |
| Apo-7 | WWKKK | –2.78 | 8 | 4.0 | 22 |
| Apo-8 | KWWKK | –2.78 | 13 | 11.4 | 99 |
| Apo-9 | KKKLI | –3.41 | <1 | n.d. | n.a. |
| Apo-10 | KKKVI | –3.94 | <1 | n.d. | n.a. |
| Apo-11 | KKWWW | –0.68 | 7 | 1.2 | 47 |
| Apo-12 | KWWWW | 1.43 | <1 | 1.3 | n.a. |
| Apo-13 | KKKKW | –4.63 | 6 | 0.8 | 9 |
| Apo-14 | KKKKK | –5.38 | <1 | 0.9 | n.a. |
| Apo-15 | RKKWF | –3.38 | 14 | 2.4 | 42 |
| cLac-BODIPY | Reference 11 | –5.49 | <1 | 4.6 | n.a. |

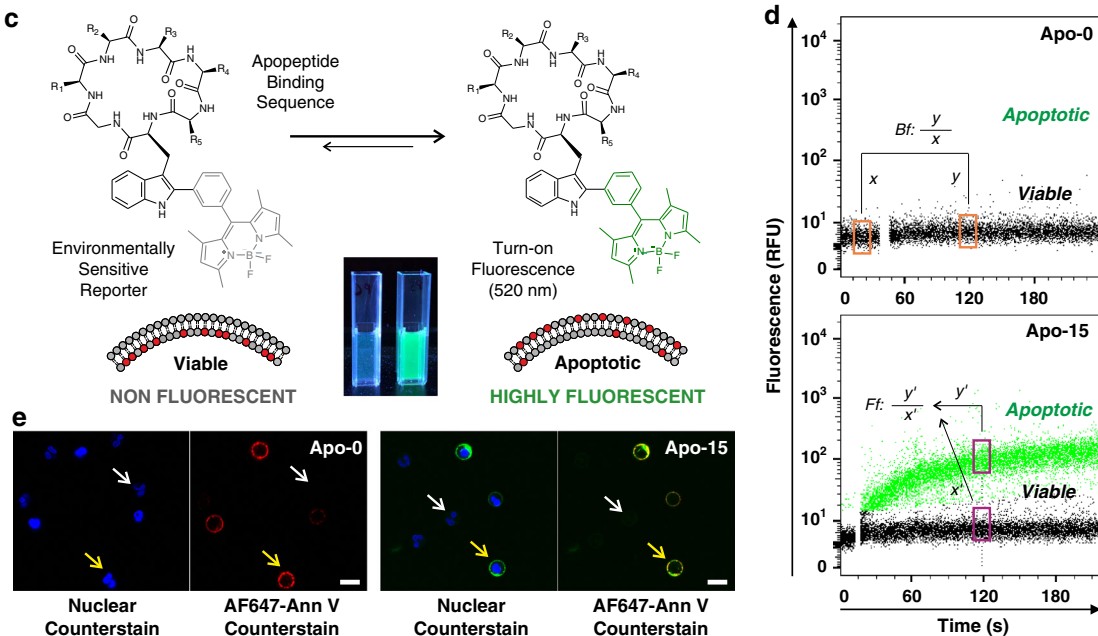

**Fig. 1 Design, synthesis, and in vitro screening of apopeptides. a** Synthetic scheme for the preparation of apopeptides (CTC-PS: 2-chlorotritylchloride polystyrene resin). Stability analysis of **Apo-15** (50 µM) under different proteolytic and oxidative environments. **b** Sequences, physicochemical and binding properties of apopeptides and cLac-BODIPY. In vitro screening of the peptides (100 nM) was performed by real-time flow cytometry in mixtures of apoptotic (AF647-Annexin V positive) and viable (AF647-Annexin V negative) neutrophils ($n \geq 3$, biologically independent experiments). **c** Fluorescence activation of apopeptides in apoptotic (right) over viable cells (left), with Trp-BODIPY as the environmentally-sensitive reporter. Pictograms of **Apo-15** under 365 nm light excitation. **d** Representative flow cytometry plots of **Apo-0** and **Apo-15** binding to mixtures of apoptotic (highlighted in green dots) and viable cells (highlighted in black dots). Fluorescence emission was recorded for around 4 min after addition of apopeptides (100 nM) without washing. Background fluorescence (Bf) was calculated as the ratio of fluorescence recorded before ($x$) and 2 min after ($y$) peptide addition. Fold increase (Ff) in apoptotic cells was calculated as a ratio of fluorescence recorded for apoptotic cells ($y'$) to that of viable cells ($x'$) after 2 min. **e** Representative fluorescence confocal microscopy images (from three independent experiments) of viable (white arrows) and apoptotic cells (yellow arrows) after incubation with **Apo-0** (left, green) and **Apo-15** (right, green) (both at 100 nM). Cells were co-stained with Hoechst 33342 (7 µM, blue) and AF647-Annexin V (5 nM, red) as nuclei and apoptosis markers, respectively ($\lambda_{exc.}$: 405, 488, 633 nm: $\lambda_{em.}$: 450, 525, 670 nm). Scale bars: 10 µm. Source data (in **b**) are provided as a Source data file.

analyses also revealed the importance of non-electrostatic interactions, with apopeptides lacking hydrophobic aromatic residues (**Apo-9**, **10**, and **14**) exhibiting poor retention of labeling. Besides, among aromatic amino acids, tryptophan increased specificity when compared with phenylalanine (**Apo-2** vs **Apo-4**).

Considering all these results, we decided to further optimize the **Apo-3** sequence (*Ff*: 14; *Bf*: 4.7; *Rt*: 47%) to reduce its background fluorescence on viable cells while retaining full binding to apoptotic cells. Arginine-rich peptides have been described to strongly bind molecular targets in other amphipathic sequences, partially due to the larger polar surface area of arginine compared with lysine[21]. Therefore, we synthesized **Apo-15** by replacing one lysine with an arginine residue. **Apo-15** [sequence: c(RKKWFW (BODIPY)G)] displays high selectivity for apoptotic cells (*Ff*: 14), marginal background fluorescence on viable cells (*Bf*: 2.4) and good retention of signal after three washes with PBS (*Rt*: 42%). Representative flow cytometry plots for **Apo-0**, which does not bind to apoptotic cells, and **Apo-15** are shown in Fig. 1d. We also examined **Apo-0** and **Apo-15** for imaging apoptotic cells in the presence of viable cells (Fig. 1e). **Apo-15** displays fluorescence in the green region of the visible spectrum, being compatible with conventional GFP and FITC filters ($\lambda_{abs.}$: 500 nm; $\lambda_{em.}$: 530 nm; Supplementary Fig. 1). Notably, **Apo-15** displays around 10-fold brightness (25,000 $M^{-1} cm^{-1}$) over other environmentally-sensitive probes for apoptosis [e.g., pSIVA (2500 $M^{-1} cm^{-1}$), N,N′-didansyl-L-cystin (3000 $M^{-1} cm^{-1}$) (Table 1)], which allows the staining of apoptotic cells at nanomolar concentrations (30–100 nM, Supplementary Fig. 2). **Apo-15** also exhibits high chemical stability under proteolytic and oxidative conditions. HPLC analysis confirmed >95% of **Apo-15** remained stable after incubation with a protease cocktail or the oxidizing agent $H_2O_2$ (Fig. 1a and Supplementary Figs. 3, 4). Altogether, these properties made **Apo-15** an optimal candidate for further characterization.

**Apo-15 delineates apoptotic cells in diverse environments**. Next, we evaluated **Apo-15** for the general detection of apoptotic cells from different species and lineages. We observed that **Apo-15** selectively stained apoptotic cells regardless of their origin. Specifically, we examined myeloid cells (neutrophils, both human and mouse, Supplementary Fig. 5), lymphoid cells (BL-2, Burkitt lymphoma) and primary epithelial cells. We performed these experiments in the presence of AF647-Annexin V to corroborate that **Apo-15** stains apoptotic and not viable cells. Notably, we observed very similar staining for **Apo-15** and AF647-Annexin V in media containing 2 mM $CaCl_2$ (Fig. 2a, b). Furthermore, **Apo-15** labeling proved to be independent of the method used to induce apoptosis [e.g., myeloid: tissue culture-induced apoptosis by culture at 37 °C for 18 h; lymphoid: irradiation with a CL-1000 Ultraviolet Crosslinker UVP at 254 nm; epithelial: treatment with staurosporine (1 μM) for 6 h], which highlights the compatibility of **Apo-15** with multiple experimental conditions.

A limitation of annexins is their dependence on high concentrations of free $Ca^{2+}$ (>1 mM), which affects their use in hypocalcemic environments in diseased tissues[22]. Therefore, we decided to assess whether **Apo-15** was able to delineate apoptotic cells independently of the concentration of free divalent cations. Notably, we observed robust binding of **Apo-15** to myeloid and lymphoid apoptotic cells in the presence of the divalent cation chelator EDTA (2.5 mM), whereas AF647-Annexin V failed to bind under the same experimental conditions (Fig. 2b–d). $Ca^{2+}$-dependent binding to apoptotic cells was also observed for polarity-sensitive annexins (pSIVA)[8] (Supplementary Fig. 6). The divalent cation-independence of **Apo-15** represents a major advantage over annexins and allows direct monitoring of apoptosis in most conditions likely to be encountered in vivo.

**Table 1 Comparative analysis of Apo-15 and commercially-available fluorescent probes for the detection of apoptosis.**

| Probe | Apo-15 | Annexins | pSIVA | N,N′-Didansyl-L-cystin | JC1 | Nonyl acridine orange | Active caspase-3/7 | PI |
|---|---|---|---|---|---|---|---|---|
| Fluorescence wavelengths ($\lambda_{exc}/\lambda_{em}$, nm) | 500/520 | Variable | 465/540 | 365/530 | 488/530–590 | 495/519 | Variable | 535/617 |
| Brightness ($\varphi \times \varepsilon$) | 25,000 | Variable | 2500 | 3000 | n.a. | 20,000 | Variable | <100 |
| Molecular weight | 1.3 kDa | ~36 kDa | ~35 kDa | 707 Da | n.a. | 473 Da | Variable | 668 Da |
| Molecular target | PS | PS | PS | n.a. | Mito memb potential | Cardiolipin | Activated caspases 3/7 | DNA |
| Early/late apoptosis | Early | Early | Early | Early | Mid | Mid-late | Late | Late |
| Divalent cation-independence | Yes | No | No | Yes | Yes | Yes | Yes | Yes |
| Wash-free imaging | Yes | High $Ca^{2+}$ | High $Ca^{2+}$ | No | No | No | No | No |
| Apoptotic cell sorting | Yes | High $Ca^{2+}$ | High $Ca^{2+}$ | n.a. | No | No | No | No |
| Functionally neutral | Yes | No | No | No | No | n.a. | No | Yes |
| Phagocytic engulfment marker | Yes | No | No | No | No | No | No | No |
| Applications | FC, IHC, CM, FP, intravital | FC, IHC, CM (high $Ca^{2+}$) | FC, IHC, CM (high $Ca^{2+}$) | FC, CM | FC, CM | FC, CM | FC, IHC | FC, IHC |
| References | This work | 4,59 | 8,60 | 19,61 | 5,62 | 63,64 | 3,65 | 66,67 |

CM confocal microscopy, FC flow cytometry, FP fluorescence polarization, IHC immunohistochemistry, PI propidium iodide.

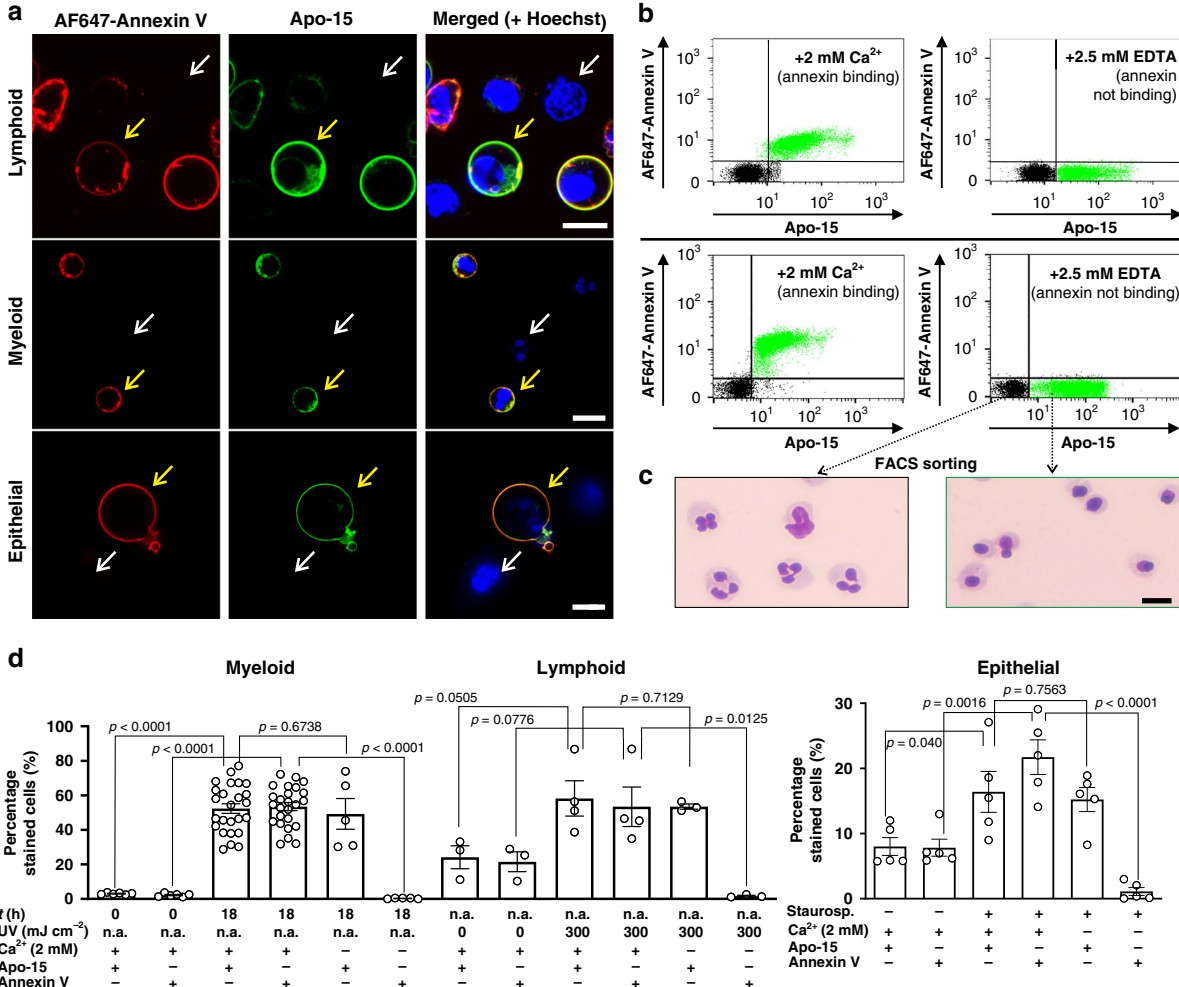

**Fig. 2 Apo-15 binds to apoptotic cells of different origin in multiple environments. a** Representative fluorescence confocal microscopy images (from three independent experiments) human apoptotic (yellow arrows) and viable (white arrows) cells from different lineages: BL-2 (lymphoid), neutrophils (myeloid), and primary airway epithelial cells (epithelial). Cells were incubated with **Apo-15** (100 nM, green), AF647-Annexin V (5 nM, red), and Hoechst 33342 (7 µM, blue) for 10 min and imaged under a fluorescence confocal microscope ($\lambda_{exc}$: 405, 488, 633 nm: $\lambda_{em}$: 450, 525, 670 nm). Scale bars: 10 µm. **b** Divalent cation-independent binding of **Apo-15** to apoptotic cells of different origin (top plots: lymphoid cells, bottom plots: myeloid cells). Mixtures of apoptotic (highlighted in green dots) and viable cells (highlighted in black dots) were stained with AF647-Annexin V (25 nM) and **Apo-15** (100 nM) in the presence of 2 mM $CaCl_2$ (left) or 2.5 mM EDTA (right). Representative histograms showing **Apo-15** binding (x-axis) vs AF647-Annexin V binding (y-axis) acquired on 5L LSR flow cytometer (n = 5). **c** Cells were sorted on their **Apo-15** positivity (green populations in panel **b**) or negativity (black populations in panel **b**). Gating strategy in Supplementary Fig. 7. Morphological analysis (four independent images from two independent experiments) of cytocentrifuge preparations of **Apo-15**-positive and **Apo-15**-negative cells under brightfield microscopy (scale bar: 10 µm). **d** Quantification of fluorescence staining of neutrophils (myeloid, $n \geq 5$), BL-2 cells (lymphoid, $n \geq 3$) and primary airway epithelial cells (epithelial, n = 5) before and after induction of apoptosis and upon treatment with **Apo-15** and AF647-Annexin V. Data acquired on 5 L LSR flow cytometer and presented as mean values ± SEM. P values obtained from two-tailed t tests. Source data (in **d**) are provided as a Source data file.

To confirm that cells stained by **Apo-15** under divalent cation-free conditions were apoptotic, we sorted a mixed population of apoptotic and viable human neutrophils (gating strategy in Supplementary Fig. 7) according to **Apo-15** labeling and examined their morphology by microscopy. **Apo-15**-positive neutrophils exhibited a pyknotic nucleus characteristic of chromatin condensation and degradation, together with cell shrinkage, which are hallmarks of apoptosis (Fig. 2c)[23]. In contrast, **Apo-15**-negative cells showed the typical multilobed nuclear morphology of viable neutrophils (Fig. 2c). The morphological appearance of **Apo-15**-positive cells was examined by scanning electron microscopy. **Apo-15**-positive neutrophils showed evidence of apoptosis-associated blebbing and pitting of the plasma membrane (Supplementary Fig. 8)[24,25]. On the other hand, the presence of microvilli-like structures, which are typical of viable cells, were

observed on the plasma membrane of **Apo-15**-negative cells (Supplementary Fig. 8). **Apo-15** also discriminated between different types of cell death by fluorescence lifetime imaging (FLIM). Human BL-2 cells were induced into apoptosis or necrosis by differential UV irradiation, incubated with **Apo-15** and imaged under a FLIM microscope to reveal that apoptotic and necrotic cells could be discriminated by their fluorescence lifetimes (Supplementary Fig. 9). Altogether, these results demonstrate that **Apo-15** is a generic marker of apoptotic cells under different physiological environments and compatible with multiple biological studies.

**Apo-15 rapidly binds PS for real-time and wash-free imaging.** In view of the specific labeling of apoptotic cells by **Apo-15**, we examined whether the amphipathic nature of **Apo-15** conferred

binding to negatively-charged phospholipids (e.g., containing PS-headgroups) that are exposed on the plasma membrane of apoptotic cells but are inaccessible in most viable cells. We used giant unilamellar vesicles (GUVs, 1–10 µm) composed of neutral phospholipids (containing only phosphatidylcholine (PC)-headgroups, 0.2 mM) or mixed with negatively-charged PS-phospholipids (PC: 0.14 mM, PS: 0.06 mM). Using fluorescence microscopy, we observed that **Apo-15** stained PC:PS GUVs with brighter intensity than PC GUVs of similar size (Fig. 3a, b). Binding to PS in PC:PS GUVs was confirmed by co-staining with the positive control AF647-Annexin V in 2 mM CaCl₂. On the

other hand, we did not observe significant differences in fluorescence intensity between PC:PS GUVs and PC GUVs when stained with the lipid marker Lissamine-Rhodamine DOPE. Furthermore, we analyzed the specificity of **Apo-15** for PS over other phospholipids [i.e., cardiolipin, PC, phosphatidylglycine, phosphatidylethanolamine (PE)] and also phosphatidic acid. **Apo-15** exhibited dose-dependent and significantly stronger binding to PS (Fig. 3b and Supplementary Fig. 10) and showed preferential binding to PS over PE, which is also known to localize on the outer leaflet of the plasma membrane following induction of apoptosis[26]. **Apo-15** proved compatible with anisotropy readings,

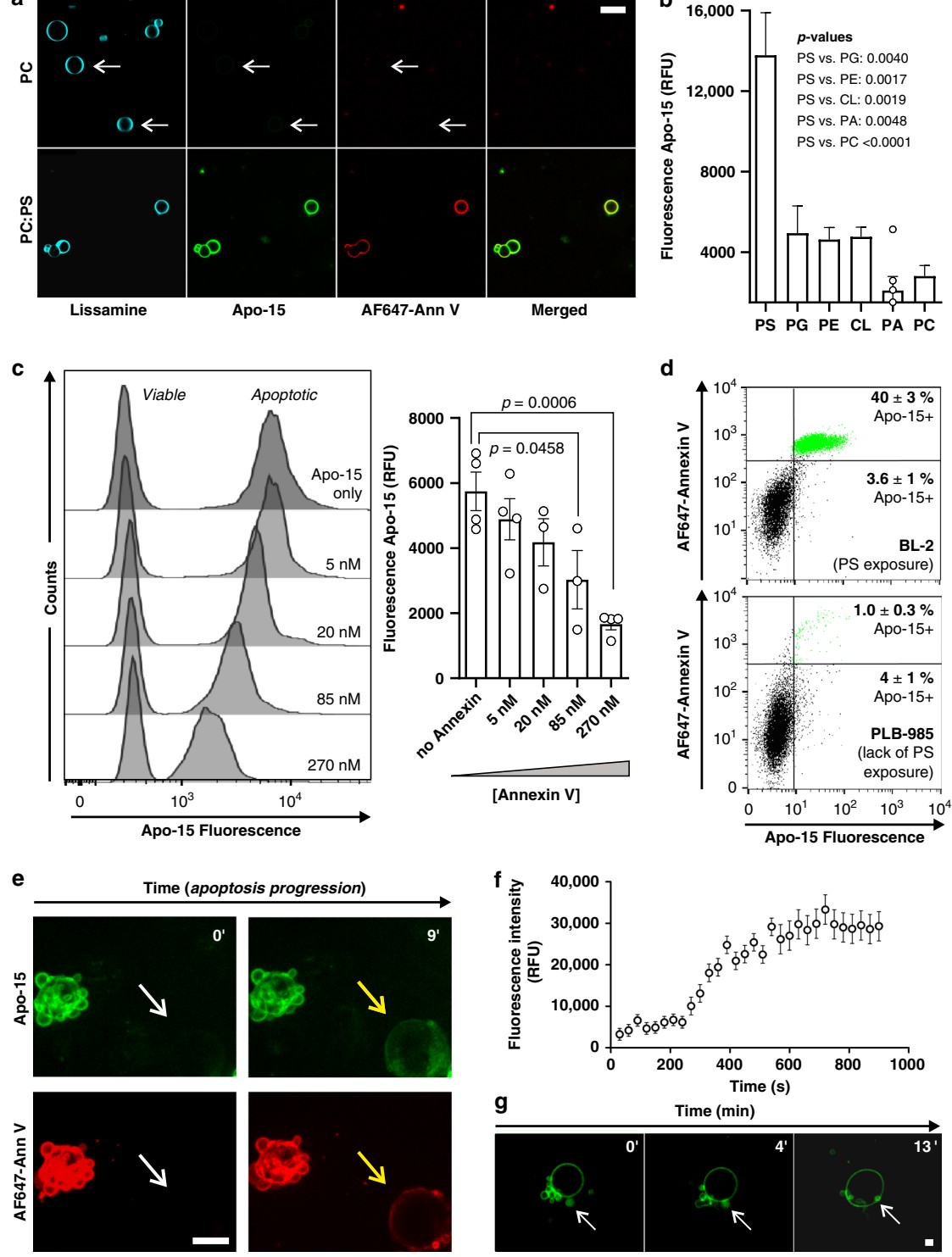

**Fig. 3 Apo-15 rapidly binds exposed PS on the surface of apoptotic cells. a** Fluorescence microscope images (three independent images from two independent experiments) of GUVs with differential lipid content (PC and PC:PS). Vesicles were stained with 0.3 mol% Lissamine-Rho-DOPE (blue), **Apo-15** (100 nM, green) and AF647-Annexin V (5 nM, red) ($\lambda_{exc.}$: 488, 561, 633 nm: $\lambda_{em.}$: 525, 593, >650 nm). Scale bar: 10 μm. **b** Binding specificity of **Apo-15** upon incubation with different phospholipids (PS: phosphatidylserine, PG: phosphatidylglycerine, PE: phosphatidylethanolamine, CL: cardiolipin, PA: phosphatidic acid, PC: phosphatidylcholine). Data acquired on a spectrophotometer ($\lambda_{exc.}$: 450 nm, $\lambda_{em.}$: 520 nm, $n > 10$, except for PA) and presented as mean values ± SEM. **c** Representative histograms showing **Apo-15** (100 nM) labeling of mixed populations of viable and apoptotic neutrophils upon competition with increasing concentrations of AF647-Annexin V, together with quantification of the mean fluorescence intensity of **Apo-15**-stained apoptotic cells ($n \geq 3$) presented as mean values ± SEM. **d** Representative histograms of BL-2 and PLB-985 cells after treatment with staurosporine (1 μM for 3 h) and staining with AF647-Annexin V and **Apo-15** ($n = 3$). **Apo-15**-stained cells are highlighted as green dots and **Apo-15**-negative cells are highlighted as black dots. Gating strategy in Supplementary Fig. 7. **e** Time-lapse fluorescence snapshots (from three independent experiments) of apoptotic BL-2 cells after UV irradiation (300 mJ cm$^{-2}$) (Supplementary Movies 1 and 2). BL-2 cells were treated with **Apo-15** (green) and AF647-Annexin V (red) and imaged every 30 s for 15 min without washing. Arrows point at a cell when viable (white) and when undergoing apoptosis (yellow). Scale bar: 10 μm. **f** Quantification of **Apo-15** fluorescence intensity and presented as mean values ± SEM over time from Supplementary Movie 1 ($n = 3$ independent measurements). **g** Time-lapse fluorescence images (from five independent cells) of subcellular apoptotic bodies (white arrow) in apoptotic neutrophils stained with **Apo-15** (green). Images were acquired every 1 s for 15 min (Supplementary Movie 3). Scale bar: 2 μm. For (**b**) and (**c**), P values were obtained from two-tailed *t* tests. Source data (in **b**, **c**, and **f**) are provided as a Source data file.

showing concentration-dependent fluorescence polarization in PS-containing liposomes due to the environmentally-sensitive nature of Trp-BODIPY (Supplementary Fig. 11).

Next, we examined whether **Apo-15** could bind PS in cell membranes. First, we performed competitive assays between AF647-Annexin V and **Apo-15** at different concentrations in co-cultures of viable and apoptotic cells. We observed that high concentrations of AF647-Annexin V could reduce binding of **Apo-15** to apoptotic cells (Fig. 3c) and also that **Apo-15** reduced Annexin V binding (Supplementary Fig. 12), suggesting that Annexin V and **Apo-15** compete for binding sites on the surface of apoptotic cells. Secondly, we used a human cell line (i.e., PLB-985) that is unable to translocate phospholipids to the outer leaflet of the plasma membrane during apoptosis due to lack of scramblase Xkr8[27]. As a control, we used human BL-2 cells, which expose PS on their membrane upon induction of apoptosis. Apoptosis in both PLB-985 and BL-2 cells was induced by treatment with staurosporine, followed by incubation with **Apo-15** and flow cytometry analysis. **Apo-15** stained around 40% of BL-2 cells, which were also stained with AF647-Annexin V, confirming that they were apoptotic (Fig. 3d). In contrast, only 1% of PLB-985 cells were **Apo-15**-positive under the same experimental conditions (Fig. 3d). We corroborated that the treatment with staurosporine induced apoptosis in PLB-985 cells by detection of cleaved caspase-3/7 (Supplementary Fig. 13)[28,29]. Taken together, these results indicate that the selective binding of **Apo-15** occurs at early stages of apoptosis when PS is translocated to the outer leaflet of apoptotic cells.

Given the rapid binding and fluorogenic properties of **Apo-15**, we evaluated its application for real-time imaging of cells actively undergoing apoptosis. BL-2 cells were induced to apoptosis by exposure to UV light and treated with **Apo-15** just before time-lapse movies were acquired under wash-free conditions in a live-cell spinning-disk microscope. We observed minimal background fluorescence suggesting that the **Apo-15** is compatible with wash-free imaging with similar signal-to-noise ratios to annexin-based reagents (Supplementary Fig. 14). We also observed cells that transitioned from being non-fluorescent (i.e., viable) to **Apo-15**-labeled (i.e., apoptotic), confirmed by simultaneous staining with AF647-Annexin V (Fig. 3e and Supplementary Movies 1 and 2). Furthermore, time-lapse quantification of the fluorescence intensity in multiple transitioning cells demonstrated that **Apo-15** bound rapidly to apoptotic cells, with more than 90% of the association occurring in <10 min (Fig. 3f). Next, we exploited these features of **Apo-15** to non-invasively image the dynamics of formation of apoptotic bodies and vesicles, which have a critical role in intercellular communication[30], and are difficult to image

with annexins due to the formation of two-dimensional crystal lattices that alter membrane dynamics[31]. Time-lapse imaging following induction of apoptosis in the presence of **Apo-15** revealed rapid release (<10 min) of subcellular vesicles and, in some cases, reintegration of vesicles back into the plasma membrane (Fig. 3g and Supplementary Movie 3). These results confirm the utility of **Apo-15** for wash-free imaging of early apoptotic events at both cellular and subcellular levels.

**Apo-15 enables imaging of apoptotic cell clearance**. Functional neutrality is an essential requirement for probes that enable imaging of biological processes under live-cell conditions. First, we tested whether **Apo-15** affected the rate at which apoptosis occurs in vitro. For these experiments, we cultured human neutrophils in the presence of different concentrations of **Apo-15** and examined the extent of tissue culture-induced apoptosis at different timepoints. Even when used at higher concentrations than those required for imaging or flow cytometry, **Apo-15** did not affect the detection or the quantification of apoptosis, as measured by staining with AF647-Annexin V (Fig. 4a).

One potential limitation to the use of annexins for imaging apoptosis is their interference with the clearance of apoptotic cells by phagocytes. This is important for in vivo imaging studies, since the perturbation of apoptotic cell clearance could result in over- or under-estimation of the extent of apoptosis within a tissue. We examined whether **Apo-15** affected the engulfment of apoptotic cells using a well-established in vitro assay to quantify the extent of phagocytosis in human monocyte-derived macrophages (MDMs) in the presence of the PS-opsonin protein S (Fig. 4b)[32,33]. Unlike Annexin V, which significantly reduces (~50%) the phagocytic capacity of MDMs, **Apo-15** did not inhibit the phagocytosis of apoptotic cells, even when it was used at high concentrations (Fig. 4c). These observations were further corroborated by fluorescence microscopy assays in co-cultures of human MDMs and apoptotic neutrophils, where we clearly observed engulfment of **Apo-15**-labeled apoptotic cells and retention of **Apo-15** staining after phagocytosis (Fig. 4d). This result opens the possibility of using **Apo-15** as a probe for tracking phagocytosis under live-cell conditions.

**Apo-15 detects drug-induced apoptosis in mouse models**. Preclinical and clinical studies have shown that low levels of free $Ca^{2+}$ are common in disease[22], with many studies (e.g., intravital imaging, in vivo therapy evaluation) being hampered by the lack of specific apoptosis reagents that are effective in those physiological environments. Given the rapid, specific and divalent

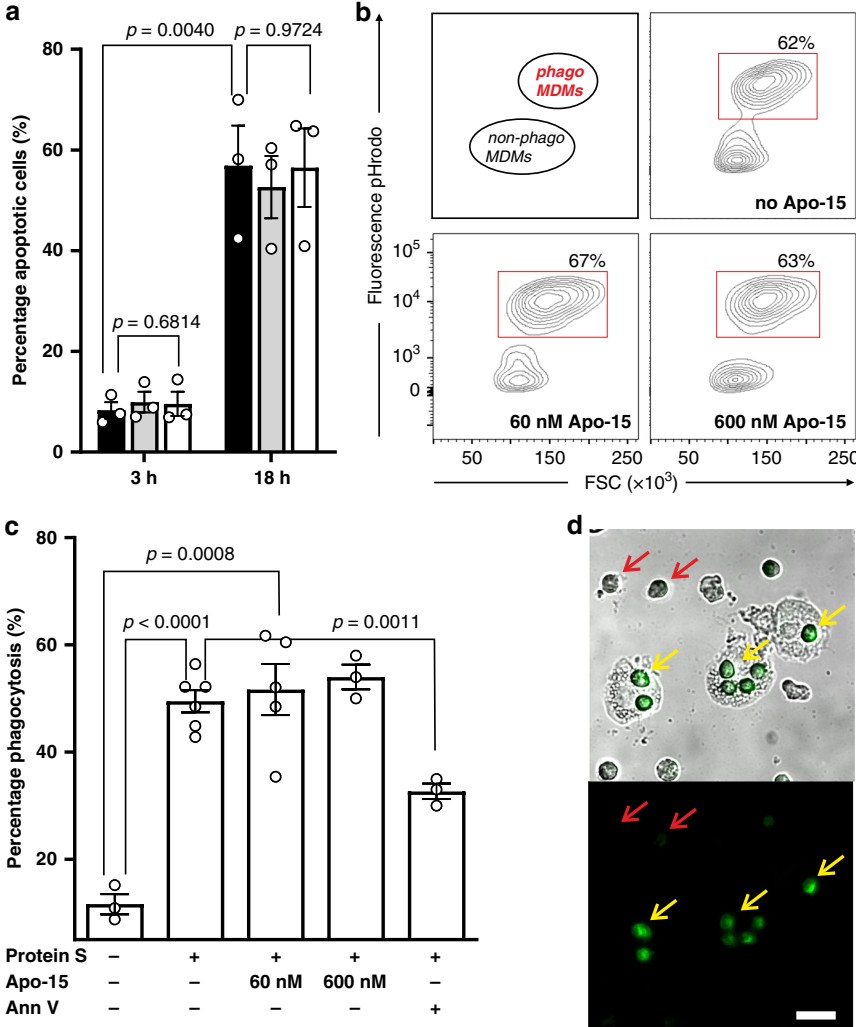

**Fig. 4 Apo-15 does not interfere with the phagocytosis of apoptotic cells. a** Percentages of human neutrophils undergoing apoptosis, as indicated by AF647-Annexin V staining, induced after culture at 37 °C for different times (3 h: left; 18 h: right). Neutrophils were cultured in the absence (black) or presence of **Apo-15** (gray: 100 nM, white: 300 nM). Data presented as mean values ± SEM ($n = 3$). **b** Representative histograms of glucocorticoid-treated human MDMs after phagocytosis of pHrodo™-labeled neutrophils in the presence of different concentrations of **Apo-15**. Cells were analyzed using a 5L LSR cytometer with MDM identified based on their distinctive forward and side scatter properties ($n = 4$). Gating strategy in Supplementary Fig. 15A. **c** Flow cytometric quantification of phagocytosis of pHrodo™-labeled neutrophils by human MDMs. Human MDMs and neutrophils were co-cultured in the absence or presence of protein S (25 nM) and **Apo-15** (60 or 600 nM) or AF647-Annexin V (25 nM). Data presented as mean values ± SEM ($n \geq 3$). **d** Representative brightfield and fluorescence microscope images (from four independent experiments) of **Apo-15**-treated human neutrophils (green) in co-culture with human MDMs. Red arrows identify **Apo-15**-negative viable neutrophils, and yellow arrows indicate MDM that have engulfed **Apo-15**-labeled apoptotic neutrophils. Scale bar: 20 μm. For (**a**) and (**c**), P values were obtained from two-tailed t tests. Source data (in **a** and **c**) are provided as a Source data file.

cation-independent binding of **Apo-15**, together with its wash-free capabilities and functional neutrality, we examined whether **Apo-15** could be used for imaging of apoptotic cells in vivo in a mouse model of ALI. This model allows temporal control of neutrophil recruitment by intratracheal administration of lipopolysaccharide (LPS), which is followed by neutrophil apoptosis and subsequent clearance by phagocytic macrophages[34]. Importantly, this is a preclinical model in which the extent of apoptosis can be modulated by cyclin-dependent kinase inhibitors (CDKi)[35]. CDKi have profound effects on inflammatory cells and some are under development as anticancer therapies[36,37].

First, we confirmed that **Apo-15** could detect CDKi-induced apoptosis in vitro. Neutrophils incubated with the CDKi *R*-roscovitine exhibited increased apoptosis by flow cytometry (as defined by Annexin V-positive staining) with a concomitant

increase in the proportion of **Apo-15** stained cells (Fig. 5a). **Apo-15** staining of CDKi-induced apoptosis was also caspase-dependent as confirmed by a significant reduction in the fluorescent signal after treatment with the caspase inhibitor QVD[34] (Fig. 5a). Next, we induced ALI in vivo in mice by intratracheal administration of *E. coli*-derived LPS (1 μg per mouse), and the apoptotic load in the lungs relative to saline control was increased by intraperitoneal administration of CDKi AT7519[38] (Fig. 5b). In all mice, we administered **Apo-15** in vivo and, 30 min later, we collected bronchoalveolar lavage fluid (BALF) together with lung tissues. BALF and lung slices from LPS-only and LPS + CDKi mice were analyzed by multiparameter flow cytometry to determine the proportion of apoptotic cells that had been stained by **Apo-15** in vivo (gating strategy in Supplementary Fig. 15). Notably, a proportion of CD45 + Ly6G+

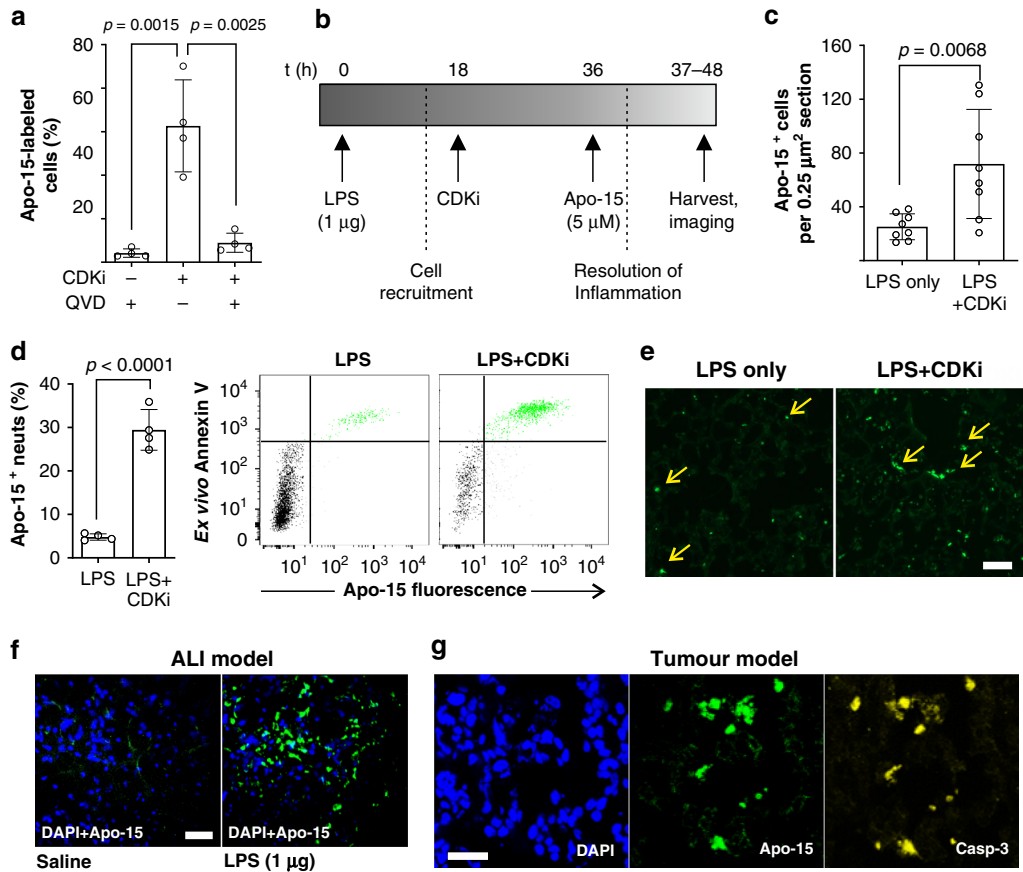

**Fig. 5 In vivo administration of Apo-15 labels drug-induced apoptosis in different mouse models. a** Human neutrophils were treated with *R*-roscovitine (20 μM) in the absence or presence of QVD (10 μM). **Apo-15**-positive cells were determined by flow cytometry (*n* = 4). **b** Experimental protocol for acute lung injury: LPS (20 μg mL⁻¹) was instilled intratracheally and mice received CDKi (30 mg kg⁻¹) or saline intraperitoneally 18 h later. **Apo-15** was administered intratracheally at 36 h. **c** Double-blind quantification of **Apo-15**-positive cells in lung tissue from mice that received LPS or LPS and CDKi (*n* = 8 per group). **d** Flow cytometric analysis of BALF from mice that received LPS (20 μg mL⁻¹) or LPS (20 μg mL⁻¹) and CDKi (AT7519, 4 mg mL⁻¹). Both groups were administered **Apo-15** (5 μM) in vivo and incubated with AF647-Annexin V (25 nM) ex vivo. Data presented as mean percentages ± s.d. of **Apo-15**-positive CD45 + LY6G+ apoptotic neutrophils (*n* = 4). Representative dot plots of CD45 + LY6G+ neutrophils with **Apo-15** positive cells highlighted in green and **Apo-15**-negative cells in black. Gating strategy in Supplementary Fig. 15B. **e** Fluorescence microscope images of sections of lung tissues from mice that received LPS or LPS and CDKi. Yellow arrows identify **Apo-15**-positive events (green) from images with *n* = 8 per group. Scale bar: 100 μm. **f** Low-magnification intravital images of lungs of saline-treated and LPS-treated mice (LPS: 20 μg mL⁻¹). **Apo-15** was administered in vivo (5 μM, green) and nuclei were counterstained with DAPI (18 μM, blue) (λ$_{exc}$: 405, 488 nm; λ$_{em}$: 450, 525 nm). Representative pictures from *n* = 3 per group. Scale bar: 50 μm. **g** High-magnification fluorescence images (from three independent experiments) of tumors from xenograft breast cancer mouse models after treatment with cisplatin (10 μg g⁻¹). **Apo-15** was administered i.v. (5 μM, green) and nuclei were counterstained with DAPI (10 μM, blue) (λ$_{exc}$: 405, 488 nm; λ$_{em}$: 450, 525 nm). Apoptotic cells were stained with anti-active caspase-3 (300 ng mL⁻¹, yellow). Scale bar: 50 μm. For (**a**), (**c**), and (**d**), *p* values were obtained from two-tailed *t* tests. Source data (in **a**, **c**, and **d**) provided as a Source data file.

neutrophils in BALF were stained with **Apo-15** (Fig. 5d). Furthermore, we quantified the effects of CDKi-induced apoptosis in vivo by measuring the percentages of apoptotic neutrophils in total cell counts. BALFs from LPS + CDKi-treated mice showed a significantly larger population of **Apo-15**-labeled cells than BALFs from LPS-treated mice (29% vs. 5%, Fig. 5d). We confirmed that CD45 + Ly6G+ cells labeled by **Apo-15** in vivo were apoptotic by co-staining with AF647-Annexin V ex vivo (Fig. 5d). Next, we acquired fluorescence microscopy images of lung tissue sections from mice under different regimes (Fig. 5c, e). We observed increased numbers of **Apo-15**-positive cells in lungs from LPS + CDKi-treated mice when compared with lungs of LPS-treated mice (90 vs 32 cells per μm² tissue, Fig. 5c) and with lungs from mice that had not been treated with **Apo-15** (Supplementary Figs. 16 and 17). Notably, the signal of **Apo-15**-labeled cells was stable following processing with 10% formalin, suggesting compatibility with fixed tissue samples. Furthermore, flow cytometric analysis demonstrated that **Apo-15**

could stain CD45 + CD11c + Annexin V− phagocytic macrophages in vivo (gating in Supplementary Fig. 15 and results in Supplementary Fig. 18). Since viable macrophages were not stained when BALFs were incubated with **Apo-15** ex vivo (Supplementary Fig. 18), we suggest that these macrophages acquire fluorescence because of internalization of **Apo-15**-labeled cells, further highlighting the potential utility of **Apo-15** for the identification of cell populations that have phagocytosed apoptotic cells.

We used **Apo-15** for intravital imaging of mice undergoing ALI using optical windows providing direct access to lungs in living mice[39]. Following direct intratracheal administration of **Apo-15**, we performed fluorescence imaging in vivo to compare the signals in the lungs of healthy and ALI mice. In healthy, saline-injected control mice, we observed negligible **Apo-15** staining whereas we observed bright stable staining in the lungs of mice undergoing ALI (Fig. 5f). Furthermore, we examined the staining of **Apo-15** following drug-induced apoptosis in the

widely used MMTV-PyMT breast cancer mouse model, which spontaneously develops tumors in the mammary glands. After tumors were grown to a size between 0.5 and 1 cm, mice were treated with cisplatin (10 µg g$^{-1}$) to induce apoptosis and **Apo-15** was injected in vivo intravenously via tail vein. As shown in Fig. 5g, the signal of **Apo-15** was retained in tumor tissues after OCT embedding and sectioning, which allowed us to compare its localization with that of active caspase-3. These results confirmed that **Apo-15** can label apoptotic cells in vivo as defined by activity of caspase-3, and that it can be administered in vivo via different routes and under various experimental conditions.

## Discussion

Dysregulation of apoptosis is a defining feature of the progression of inflammatory diseases and of cancer. Although a number of probes for the detection of apoptosis using non-optical imaging modalities (e.g., positron emission tomography, single-photon emission computed tomography) have been described[40,41], there are very few optical probes that can be used in vivo (Table 1). In particular, chemical degradation in an inflammatory microenvironment (e.g., protease or oxidative), dependence on in vivo availability of high concentrations of free divalent cations and perturbation of cellular functions represent key limitations.

We have rationally designed and synthesized fluorogenic heptapeptides that specifically overcome these limitations, allowing us to identify a highly stable optical imaging reagent for apoptotic cells (termed **Apo-15**). The functional neutrality of **Apo-15**, together with the characterization of its binding to apoptotic cells in vitro and in vivo, suggests that **Apo-15** can be used for non-invasive imaging of apoptosis in a broad range of experimental settings, including different preclinical mouse models.

First, we have established a solid-phase synthetic method for the production of apopeptides (including **Apo-15**) as low molecular weight (~1 kDa) fluorogenic agents for the detection of PS, widely regarded as an early biomarker of apoptosis. **Apo-15** is shorter than other peptides reported to bind phospholipids related to apoptosis (e.g., PE-binding duramycin and PS-binding PSBP-6 are 19-mer and 14-mer peptides, respectively)[40]. Furthermore, **Apo-15** incorporates Trp-BODIPY as an environmentally-sensitive reporter for exclusive emission in proximity to cellular membranes with enhanced brightness over other fluorophores (Table 1). **Apo-15** can be produced cost-effectively and in large scale (>100 mgs/ batch, which would allow ~300,000 in vivo imaging assays). In contrast, PS-binding proteins like annexins (~36 kDa) or the Ca$^{2+}$-independent lactadherin C2 (~17 kDa) are difficult to manufacture with site-specific chemical tags in large scale. The small size and peptidic nature of **Apo-15** might also favor better clearance and biodistribution to tissues where proteins (e.g., annexins) cannot gain access. For example, **Apo-15** shares structural similarities (e.g., cyclic core, positively-charged amino acids) with recently developed blood brain barrier shuttle peptides[42,43], suggesting that **Apo-15** or slightly modified derivatives could enable optical detection of apoptotic cells in the brain.

Second, we have demonstrated that **Apo-15** binds rapidly to aminophospholipids, with a strong preference for PS over other phospholipids. Of note, **Apo-15** binds more strongly to PS than PE. This observation suggests that the binding to PS may be favored by electrostatic interactions between the positively-charged residues (i.e., arginine, lysine) of **Apo-15** and the free negatively-charged carboxylate groups in PS, which are not present in PE. We also showed that **Apo-15** binds to cells from different species and lineages undergoing tissue culture-induced apoptosis as well as chemical or UV-induced apoptosis (Fig. 2). Interestingly, although our data demonstrate that Annexin V and

**Apo-15** cross-compete for binding (Fig. 3), fluorescence analysis reveals that dual labeling with these two probes is not strictly co-linear. One possibility is that there are differences in lipid specificity between Annexin V and **Apo-15**. Alternatively, the assembly of Annexin V multimers on the apoptotic cell membrane[31] may restrict accessibility to the lipids. Detection of membrane lipids offers key advantages (e.g., no need for fixatives) over optical agents visualizing other events (i.e., caspase activation, mitochondrial dysfunction or loss of membrane integrity; Table 1) and allows identification of apoptotic cells at early stages of disease where therapeutic interventions may exert beneficial effects.

Third, the divalent cation-independence of **Apo-15** binding enables in vivo detection and quantification of apoptosis in tissues (e.g., lungs, Fig. 5) where low and/or variable levels of free Ca$^{2+}$ could limit the performance of annexin-based reagents. **Apo-15** is also compatible with in vitro experimental conditions where the use of annexins would be challenging, such as assays where calcium ion concentrations need to be tightly controlled due to potential activation of cells (e.g., the binding of platelets to fibrinogen is highly dependent on the concentration of extracellular free calcium)[44]. In addition, the enhanced fluorescence emission of **Apo-15** upon binding allows time-lapse imaging of apoptosis progression in situ (Supplementary Movie 1) without the need for washing and without significant loss of signal over time. We have exploited these characteristics to visualize the different stages in the formation and reabsorption of apoptotic bodies, which will provide insights into the functions that these structures play in different diseases (Fig. 3g and Supplementary Movie 3).

Fourth, we have demonstrated that **Apo-15** is a highly stable peptide, even under proteolytic or oxidative conditions (Fig. 1), overcoming an important limitation of protein-based agents. The low molecular weight of **Apo-15** (~27-fold smaller than annexins) may allow increased tissue penetration and biodistribution in vivo. This feature could also enable imaging studies in optically-transparent organisms that are well-established for developmental studies (e.g., zebrafish (*D. rerio*), worms (*C. elegans*), flies (*D. melanogaster*)), given that apoptosis is crucial for the appropriate formation of organs and structures during embryonic development[45]. We have also demonstrated that the staining capabilities of **Apo-15** are compatible with different administration routes. **Apo-15** could facilitate mechanistic and efficacy studies of drugs in tissues undergoing repair and/or regeneration (e.g., liver). There is experimental evidence that the first phases of liver injury are characterized by hepatocyte apoptosis, which is profibrogenic[46]. With drugs modulating apoptosis currently being tested for the treatment of liver injury, the repeated in vivo administration of **Apo-15**—which may be prohibitive in the case of annexins due to its cost—could represent a powerful method for longitudinal monitoring of hepatocyte apoptosis in the liver using abdominal intravital imaging windows[47].

Finally, we and others have previously shown that promotion or inhibition of apoptosis affects the progression and resolution of inflammation[48]. In this work, we have demonstrated that **Apo-15** does not accelerate or delay apoptosis, a critical requirement for quantitative assessment of cell death in vivo. In contrast to annexins and lactadherin C2 proteins[33,49], **Apo-15** does not inhibit the phagocytic removal of apoptotic cells by macrophages (Fig. 4), and therefore represents a valuable tool for imaging assays where phagocytosis and cell clearance must remain unaffected. This functional neutrality of **Apo-15** may enable tracking of the ultimate fate of apoptotic material in mice. Further studies (we were unable to co-stain ex vivo samples with **Apo-15** and antibodies given that processing and antigen retrieval methods

results in marked loss of the **Apo-15** signal) are required to investigate whether pulses of **Apo-15**-labeled apoptotic cells could be followed in vivo. Also, further studies would be required to determine the cell types that are preferentially labeled in vivo and the longevity of the **Apo-15** fluorescence signal, particularly when internalized apoptotic material reaches the acidic phago-lysosome. Given experimental evidence that links dysfunctional phagocytosis with worsening of pulmonary diseases (e.g., COPD, cystic fibrosis)[50], **Apo-15** could open avenues for imaging of apoptosis in other preclinical models and in clinical specimens. Accumulation of apoptotic cells has been also reported in atherosclerotic lesions, where defective phagocytic clearance can contribute to the establishment of a pro-inflammatory necrotic core and disease progression[51]. In vivo imaging of apoptosis in these lesions, now generally done via positron emission tomography (PET), could allow the identification of at risk vulnerable plaques with pro-thrombotic potential[52]. The Trp-BODIPY in **Apo-15** contains two $^{19}F$ fluorine atoms which can be isotopically exchanged with radioactive $^{18}F$ to prepare radiolabeled **Apo-15** analogs for multimodal imaging (i.e., PET and optical imaging)[53]. This approach could help to identify vulnerable plaques in vivo and ex vivo using a single molecular agent, and also allow us to test the efficacy of therapies promoting efferocytosis and reducing apoptotic cell burden within atherosclerotic lesions.

In summary, our studies validate **Apo-15** as a reliable fluorogenic peptide for the detection and quantification of apoptosis, both in vitro in multiple cell lineages and in vivo in different preclinical mouse models of inflammation and cancer. The specificity and non-invasiveness of **Apo-15** could allow imaging of apoptotic events that are not possible with current probes, from tracking of apoptotic bodies with subcellular resolution to performing intravital imaging using optical windows. Moreover, **Apo-15** labeling provides a simple and effective readout to directly assess the efficacy of therapeutic interventions targeting apoptosis in vivo.

## Methods

**Chemical synthesis and characterization**. The synthesis and analytical characterization data for apopeptides is described in the Supporting Information (Supplementary Tables 1-2 and Supplementary Methods).

**Cell culture and in vitro induction of apoptosis**. Work with human peripheral blood leukocytes complied with all relevant ethical regulations and informed consent was obtained. The study protocol was approved by the Accredited Medical Regional Ethics Committee (AMREC, reference number 15-HV-013) at the University of Edinburgh. Human blood leukocytes were isolated from healthy volunteers[54]. Briefly, whole blood was anti-coagulated with sodium citrate [0.4% (w/v) final concentration] and centrifuged at $350 \times g$ for 20 min. Platelet-rich plasma was removed and autologous serum was generated by recalcification with CaCl₂ (20 mM final concentration). Erythrocytes were separated from leukocytes by sedimentation with Dextran T500 (0.6%) for 30 min and leukocyte populations were further fractionated using isotonic Percoll (GE Healthcare) 49.5%/63%/72.9% discontinuous gradients. Polymorphonuclear cells (>95% neutrophils) were harvested from the 63%/72.9% interface. Neutrophil apoptosis was induced by in vitro culture for 18 h at 37 °C and 5% CO₂ in IMDM supplemented with 5% autologous serum. Monocytes were enriched from the mononuclear cells (49.5%/63% interface) by negative selection using the Pan Monocyte Isolation Kit (Milentyi Biotech) according to the manufacturer's instructions. Monocytes were cultured for 6 days in IMDM supplemented with 5% autologous serum to yield monocyte-derived macrophages (MDMs).

For work with human cell lines, BL-2 and PLB-985 cells were cultured in RPMI supplemented with 10% FBS, 100 U mL⁻¹ penicillin and 100 µg mL⁻¹ streptomycin. Cells were plated on 8-well confocal chambers ($1.25 \times 10^5$ cells/well) the day before imaging experiments. Apoptosis was induced by UV treatment (300 mJ cm⁻² with intervals of $6 \times 50$ mJ cm⁻² every 30 s in BL-2 cells) or by chemical treatment (staurosporine 1 µM for 3 h in BL-2 and PLB-985; staurosporine 1 µM for 6 h in primary airway epithelial cells).

Primary Small Airway Epithelial cells (HSAEC) were cultured in airway epithelial cell basal media supplemented with Bronchial Epithelial Growth Kit. Cells were plated on 8-well confocal chamber ($5 \times 10^4$ cells/well) the day before imaging experiments. For flow cytometry experiments, cells were detached using

0.05% trypsin-EDTA for 3 min at 37 °C and resuspended in HEPES-NaCl buffer containing 0.1% BSA and 2 mM CaCl₂ or 2.5 mM EDTA.

Mouse bone marrow-derived neutrophils were purified from C57Bl/6 mice using discontinuous Percoll gradients as previously described[55]. The purity of the neutrophil population was determined as 75–80% by analysis of cytocentrifuge preparations. Neutrophils were induced to undergo tissue culture-induced apoptosis by culturing for 18 h at 37 °C and 5% CO₂ in IMDM with 10% FBS.

**Flow cytometry and fluorescence-activated cell sorting**. All apopeptides were reconstituted at 5 mM in DMSO. Peptides were used at 100 nM in HEPES-NaCl buffer containing 2 mM CaCl₂, with or without 2.5 mM EDTA, and were incubated with the cells for 10 min at r.t. before flow cytometry analysis, unless otherwise stated. For co-labeling experiments, AF647-Annexin V and PI were used at the indicated concentrations, with AF647-Annexin V being added to cells 10 min prior to analysis and propidium iodide being added directly before analysis. BALF analysis: $10^6$ cells were stained for 30 min on ice with anti-mCD45-PE, anti-mLy6G-PerCpCy5.5, and anti-mCD11c-Pacific Blue diluted 1:100 in HEPES-NaCl buffer. After washing, AF647-Annexin V was added at 25 nM in HEPES-NaCl buffer containing 2 mM CaCl₂ for 15 min prior to acquisition. Fluorescence emission was measured on a 5 L LSR flow cytometer (BD). Excitation sources/emission filters used: apopeptides (488 nm/525 nm); AF647-Annexin V (641 nm/670 nm); PI (560 nm/600 nm), anti-mCD45-PE (560 nm/580 nm), anti-mLy6G-PerCP-Cy5.5 (488 nm/700 nm), anti-mCD11c-Pacific Blue (405 nm/450 nm). Cells were sorted on the FACS Aria (BD), and data were analyzed with FlowJo software.

**Time-lapse microscopy and live-cell imaging**. Time-lapse microscopy of apoptosis in BL-2 cells was performed using a spinning disk confocal microscope (Andor) equipped with a live-cell incubation system (37 °C, 5% CO₂). Live-cell imaging of different lineages was performed on a confocal laser scanning microscope TCS SP8 (Leica) at 37 °C and 5% CO₂. Prior to imaging, freshly isolated neutrophils were incubated for 3 h at 37 °C in phenol red-free IMDM (Gibco) supplemented with 2.5% autologous serum and 20 µM R-roscovitine (Selleckchem). UV-treated BL-2 cells were incubated for 3 h at 37 °C prior to imaging. AF647-Annexin V and Hoechst 33342 were used as counterstains at the indicated concentrations. Apopeptide staining was performed by direct addition of apopeptides at 100 nM and imaged after 10 min. Images were processed with ImageJ software.

Signal-to-noise ratios for apoptotic neutrophils labeled with FITC-Annexin V or **Apo-15** were determined after image acquisition under the same laser power and gain and analysis using ImageJ. Briefly, raw images were converted into 16-bit grayscale images and average fluorescence intensities in labeled cells and in the background were measured (≥150 points per condition). Experiments were performed in triplicate.

**Liposome-based assays**. Giant unilamellar vesicles (GUV) were prepared by the electroformation method[56]. Briefly, stock solutions of lipids (0.2 mM total lipid containing 0.3 mol% Lissamine-Rho-DOPE) were prepared in CHCl₃:MeOH (2:1, v/v) and deposited on Pt wires that were placed under vacuum for 2 h to completely remove the organic solvents. 500 µL assay buffer (25 mM HEPES, 150 mM NaCl, pH 7.4) pre-warmed at 37 °C was then added to the chamber, and the Pt wires were connected to a TG330 function generator (Thurlby Thandar Instruments), with the field being applied in three steps at 37 °C: (1) frequency 500 Hz, amplitude 260 mV (29 V m⁻¹) for 6 min; (2) frequency 500 Hz, amplitude 3.1 V (320 V m⁻¹) for 20 min; (3) frequency 500 Hz, amplitude 7.3 V (656 V m⁻¹) for 90 min. GUVs were visualized under an inverted confocal fluorescence microscope (Nikon D-ECLIPSE C1) with the following excitation wavelengths: 488 nm for **Apo-15**, 561 nm for Lissamine-Rho-DOPE, and 637 nm for AF647-Annexin V. Images were collected using band-pass filters (515 ± 15 nm for **Apo-15**, 593 ± 20 nm for Lissamine-Rho-DOPE), and a long-pass filter >650 nm for AF647-Annexin V. For AF647-Annexin V staining, buffer containing 25 mM HEPES, 150 mM NaCl, 2 mM CaCl₂, and 0.1% BSA was used. Polarized fluorescence emission was measured in a fluorimeter plate reader PHERAstar (BMG) containing polarizers and filters with excitation at 485 nm and emission at 520 nm.

**Binding of Apo-15 to phospholipids**. Cardiolipidin, PG, and PA were purchased from Stratech Scientific Ltd. PS and PC were obtained from Sigma-Aldrich and PE was purchased from Cambridge Biosciences. All lipids were dissolved at 1 mg mL⁻¹ in dry EtOH and transferred into a black flat-bottom 96-well plate to generate lipid thin layers by solvent evaporation at r.t. **Apo-15** was reconstituted at 1 µM in sterile PBS and incubated at 25 °C for 40 min. Fluorescence emission was measured in a Synergy H1 BioTek spectrophotometer ($\lambda_{exc}$: 450 nm, $\lambda_{exc}$: 520 nm). All experiments were performed in triplicate.

**Competition binding assays between Apo-15 and AF647-Annexin V**. Neutrophils which had been induced to undergo tissue culture-induced apoptosis (cultured for 18 h at 37 °C and 5% CO₂ in IMDM supplemented with 5% autologous serum) were pre-incubated for 10 min with different concentrations of AF647-Annexin V or **Apo-15** (as indicated in Fig. 3 and Supplementary Fig. 12) in HEPES-NaCl buffer containing 0.1% BSA and 2 mM CaCl₂. Pre-incubation was

followed by addition of AF647-Annexin V or **Apo-15** at the indicated concentrations in HEPES-NaCl buffer containing 0.1% BSA and 2 mM CaCl₂. Histograms and mean fluorescence intensities of apoptotic cells were measured on a 5L LSR flow cytometer under the following excitation/emission filters: **Apo-15** (488 nm/525 nm), AF647-Annexin V (641 nm/670 nm). Data were acquired using the FACS Diva software and analyzed using the FlowJo X software.

**Phagocytosis assays**. To ensure protein-free conditions in the phagocytosis assay, cells were washed with Hank's balanced salt solution without Ca²⁺ or Mg²⁺ containing 2.5 mM EDTA. Prior to the phagocytosis assay, aged neutrophils (18 h culture at 37 °C) were labeled with pHrodo^TM Red SE (ThermoFisher). To examine uptake of apoptotic neutrophils in the presence and absence of PS opsonins, the serum component, protein S (Enzyme Research Labs) was added to a final concentration of 25 nM. MDMs were then co-incubated for 40 min with apoptotic neutrophils at 37 °C at a phagocyte:target ratio of 1:6. Non-ingested apoptotic neutrophils were removed by aspiration and cells were detached with 0.05% trypsin-EDTA for 5 min at 37 °C followed by incubation on ice. The percentages of fluorescent MDMs (i.e., those that had ingested apoptotic cells) were determined on the 5L LSR (BD Bioscience). For imaging, MDM was cultured on 9-mm-diameter round glass coverslips (ThermoFisher) and mounted with ProLong gold (ThermoFisher) on microscope slides. Images were taken on a LSM 510 Zeiss confocal microscope.

**In vivo monitoring of apoptosis in a model of lung inflammation**. Animal testing and research complied with all relevant ethical regulations. The study protocol was approved by the UK Home Office at the University of Edinburgh and by the University of Edinburgh animal welfare Committee (Project License: PPL 60/4502). Mice were housed in a specific-pathogen-free facility with standard husbandry. In a mouse model of acute lung inflammation[34], 8–12-week-old C57BL/6 mice were administered intratracheally E. coli-derived (O127:B8) LPS (20 μg mL⁻¹, 50 μL sterile saline per mouse), combined S. aureus-derived LTA (150 mg) and PepG (50 mg). Dark and light cycles were 12 h. After 18 h, mice received either 30 mg kg⁻¹ AT7519 intraperitoneally in 200 μL sterile saline or 200 μL of vehicle control. At 36 h post-LPS instillation, 5 μM **Apo-15** in 50 μL sterile saline was administered, and lavages and lung tissue were harvested 30 min later. Lungs were lavaged with three aliquots of 800 μL sterile ice-cold saline. Bronchoalveolar lavage fluid (BALF) was centrifuged at 300 × g for 5 min and BALF cells were analyzed for flow cytometry and Diff-Quick staining. Lungs for histological analysis were perfused with saline followed by fixation with 10% formalin.

**Intravital imaging**. Female C57BL/6 mice at 8 weeks of age were administered E. coli-derived LPS (20 μg mL⁻¹, 50 μL sterile saline per mouse) intratracheally or an equal volume of saline. After 48 h, 50 μL of **Apo-15** (5 μM) in saline (<0.1% DMSO) was administered intratracheally. Mice were anesthetized by intraperitoneal injection of ketamine/dexdomitor (100 mg/kg/0.5 mg/kg). Saline (500 μL) was injected peritoneal prior to surgery, tracheotomies performed, and mechanical ventilation (MiniVent Model 845, Harvard Apparatus) initiated with the mice on a heated stage. Thereafter, mice were kept anesthetized at 0.8–1.0% isoflurane delivered through the ventilator at 100 breaths per minutes with 25 mL H₂O maintaining positive end expiratory pressure. A thoracotomy was performed[39]. Briefly, an incision was made between the 4th and 5th left ribs and a 3D-printed thoracic suction window with 8-mm coverslip inserted. The window was kept in place with single-axis translator micromanipulators (#MSH1.5 Thor Labs), and vacuum pressure (15–20 mm Hg) was applied to avoid motion artefacts of the imaged tissue regions. At the time of imaging, 50 μL of DAPI (5 μg mL⁻¹) was administered intravenously and images captured for up to 1–2 h. Mice were imaged with a custom-built spinning disk confocal microscope (Solamere Technologies) modified to upright configuration using a LD 20X Fluar 0.4 NA lens (Zeiss Microscopy)[57]. The microscope was controlled by MicroManager software and time-lapse movies generated from the acquired images using Imaris (Bitplane) software.

**In vivo staining of cisplatin-induced apoptosis in a MMTV-PyMT breast cancer mouse model**. Animal testing and research complied with all relevant ethical regulations. The study protocol was approved by the Institutional Animal Care and Use Committee (IACUC) at Cold Spring Harbor Laboratories. Animal experiments were conducted in accordance with the NIH Guide for Care and Use of Laboratory Animals. MMTV-PyMT (C57BL/6 background) is a reported breast cancer mouse model to spontaneously develop tumors in the mammary glands as previously described[58]. Tumors were grown to a size between 0.5 and 1 cm, and then mice were treated with cisplatin (10 μg g⁻¹) to induce apoptosis.

Prior to administration of **Apo-15**, mice were anaesthetized with 4% isofluorane and kept anesthetized under 1–2% isoflurane and 21% oxygen and balanced with nitrogen at 1 L min⁻¹. Mice were given **Apo-15** in vivo intravenously (5 μM, total volume 100 μL) and tumor tissues were harvested after 2 h. Tissues were fixed in 4% paraformaldehyde for 2 h, incubated in gradual increasing concentrations of sucrose, embedded in OCT media and frozen. Tumor tissues were then sectioned, rehydrated, and blocked with PBS containing 2.5% BSA and 20% goat serum. Primary antibody staining was performed overnight with 1:200 rabbit anti-cleaved

caspase-3 antibody (Cell Signalling) in PBS containing 1.25% BSA and 10% goat serum at 4 °C. Slides were then washed three times with PBS and stained with 1:400 secondary goat anti-rabbit AF647-IgG (H + L) and DAPI (10 μM) in PBS containing 1.25% BSA and 10% goat serum for 1 h at r.t. Slides were mounted using ProlongGold, imaged under a Leica SP8 confocal microscope. Excitation/emission filters used: DAPI (405 nm/450 nm), **Apo-15** (488 nm/525 nm), and AF647 (641 nm/670 nm). Images were analyzed and processed with ImageJ software.

**Statistical analysis**. Statistical differences were analyzed two-sided unpaired t tests using Graphpad software. Independent experiments were performed at least three times, unless otherwise mentioned.

**Reporting summary**. Further information on research design is available in the Nature Research Reporting Summary linked to this article.

## Data availability

The source data underlying Figs. 1b, 2d, 3b, c, f, 4a, c, 5a, c and d and Supplementary Figs. 1, 4, 10, 11, 12 and 14 are provided as a Source data file. Additional data that support findings of this study are available from the corresponding authors upon reasonable request. Source data are provided with this paper.

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

## Acknowledgements

N.D.B. acknowledges funding from OPTIMA (EP/L016559/1). R.S.F. acknowledges an MSCA Individual Fellowship (659046). L.M.T. acknowledges the support of Fundacion Antonio Martin Escudero (FAME) in the form of a post-doctoral fellowship. S.T.H. is an Australian Research Council Future Fellow (FT150100398). R.L. acknowledges funding from MINECO (CTQ2015-67870-P and ERA NET PCIN-2015-224). F.M.G. acknowledges financial help from the Spanish Ministry of Economy (FEDER MINECO PGC2018-099857-B-I00) and the Basque Government (IT264-19). M.E. acknowledges funding from Cold Spring Harbor Laboratory Cancer Center (P30-CA045508), North-well Health, and the Thompson Family Foundation. M.V. acknowledges funding from an ERC Consolidator Grant (771443), the Biotechnology and Biological Sciences Research Council (BB/M025160/1), and The Royal Society (RG160289). The authors thank the technical support from the QMRI Flow Cytometry and Confocal Advanced Light Microscopy facilities at the University of Edinburgh and Yaiza Varela (UPV), Lux-embourg Bio Technologies Ltd. (Rehovot) for the kind supply of reagents for peptide synthesis, and Astex Therapeutics, who kindly provided AT7519 as a gift. The authors also thank Prof. Chris Gregory (University of Edinburgh) for provision of BL-2 cells and valuable discussions.

## Author contributions

N.D.B. and R.S.F. synthesized chemical compounds and performed in vitro characterization experiments; L.M.T. and R.L. contributed to chemical syntheses and chemical characterization; N.D.B., R.D., J.A.C., A.G.R., M.A.S., and M.E. designed and performed in vivo experiments; J.S., F.G., S.T.H., and M.V. designed and performed in vitro experiments in liposomes; J.A.M. and S.V. provided cell lines and assisted with in vitro experiments; N.D.B., R.S.F., I.D., and M.V. analyzed the data and discussed the results; I.D. and M.V. conceived and co-supervised the overall project and wrote the paper with contributions from all authors.

## Competing interests

M.V. declares competing interests as The University of Edinburgh signed a licensing agreement with BioLegend for the commercialization of **Apo-15**. The remaining authors declare no competing interests.
