## [Peer Review File · Nature Communications]

Reviewers' comments:

Reviewer #1 (Remarks to the Author):

In this manuscript, Barth et al. developed a cyclic peptide that binds phosphatidylserine (PtdSer) for the in vivo imaging of apoptotic cells. I have several comments.

General Comments:

1. The author's group previously published a similar cyclic peptide (called cLac-Bodipy) that was claimed specifically bind to negatively charged phospholipids including PtdSer. How apo-peptides are different from cLac-Bodipy?
2. Biochemical characterization of apo-peptides is not sufficient. More detailed binding specificity to the phospholipids should be shown at least for Apo-15.
3. Several other groups designed PtdSer-binding peptides, but due to the insufficient affinity and specificity, none of them are successful for imaging apoptotic cells in vivo (for review, see please see Rybczynska et al, *Med. Res. Rev.* 38, 1713, 2018.). Do the authors think that Apo-15 is better than others? Please discuss with more data (specificity and affinity).

Detailed Comments

1. Figure 1B: Please include cLac-Bodipy in the Table for comparison. Please describe how the sequence of the 15 apo-peptides is designed.
2. Figure 1C: The idea of Bodipy was explained in the authors' own previous publications (reference 10-11). This Figure may not be necessary.
3. In the cell death field, APO-1 is known as a cell death receptor (Fas) (*Cell Death Differ.* 10, 26, 2013).
4. Figure 2A: The authors claim that binding of Annexin V and Apo-15 to PtdSer is not competitive. What kinds of mechanism the authors propose? Annexin V and Apo-15 bind to different parts of PtdSer? In the middle panel of Figure 2, I can find a cell (at left) that is strongly bound by Annexin V, but only weakly bound by Apo-15, while the cell at middle has an opposite relation. I want to see the effect of high concentration of Annexin V on the Apo-15 binding.
5. Figure 2A legend: Why the concentration of Apo-15 (100 nM) and Annexin (0.01 µg/ml) are described in different ways (molar or mass concentration)? The molar concentration of 0.01 µg/ml Annexin is about 0.3 nM, indicating that the affinity of Apo-15 to apoptotic cells is 100 times weaker than that of Annexin. Yes?

6. Figure 2C: If Annexin V and Apo-15 similarly bind to PtdSer on apoptotic cells, why the binding of Apo-15 to apoptotic cells is not proportional to that of Annexin V?
7. Figure 3B: I want to know whether Apo-15 binds to other phospholipids (cardiolipin, phosphatidylglycine, phosphatidylethanolamine, phosphatidylinositol, and phosphatic acids) or not. How is K_d (Dissociation Constant) of Apo-15 to PtdSer?
8. Figure 3C: Scramblases that are activated during apoptosis are non-specific for phospholipids. Thus, apoptotic cells expose not only PtdSer, but also other phospholipids such as phosphatidylethanolamine and phosphatidylinositol. Since apoptotic scramblases are not expressed in PLB-985 cells, not only PtdSer but also many other phospholipids (mentioned above) will not be exposed.
9. Figure 3C and 3D: According to Fig. 3D, about 20% of staurosporine-treated PLB are apoptotic. In Figure 3C, there is a population that is recognized by APO-15 but not by Annexin V. What is this population? They are not apoptotic? In Reference 7, binding of MFG-E8 to apoptotic cells was shown by double staining with MFG-E8 and TUNEL (DNA fragmentation). Is a similar approach not possible?
10. Figure 4B: It is well known that viable cells expose PtdSer under some circumstances. I am not sure what the authors want to say in this Figure. Do the authors want to say that Annexin V is much more sensitive than Apo-15 to detect PtdSer, or Annexin V binds non-specifically to viable cells? Please show the binding profiles of Annexin V and Apo-15 to apoptotic monocytes. Do they bind to apoptotic monocytes to the same extent?
11. Figure 4C and D: Please describe how much Apo-15 and Annexin V were added to the phagocytosis assay.
12. Figure 4: The authors claim that Annexin V inhibits the engulfment of apoptotic cells, but Apo-15 does not. This is misleading. A large amount of PtdSer is exposed on the surface of apoptotic cells. If most of the PtdSer is masked by Annexin V, the engulfment will be inhibited. Thus, the inhibition by Annexin V is dose-dependent. Similar dose-dependent inhibition of phagocytosis was shown with other PtdSer-binding protein (Reference 7). The concentration of Apo-15 used to inhibit the engulfment is the concentration at which the binding of Apo-15 to PtdSer is saturated? Do the authors want to say that the binding site of Apo-1 on PtdSer is different from that recognized by macrophage (or Protein S) for engulfment? Otherwise, it seems that Apo-15 recognizes a molecule rather than PtdSer so that PtdSer can work as an "eat me" signal.
13. Page 3, line 55: Reference 7 is not a correct reference here.
14. The authors compare Apo-15 with Annexin V, and claim that different from Annexin V, it does not require Ca²⁺. It should be noted that Lactadherin (MFG-E8) C2 does not require Ca²⁺, too.
15. Figure 5: The authors claim that Apo-15 can be used to detect apoptotic cells in vivo. But, technetium-labelled C2 domain of lactoadherin or synaptotagmin and I124 labeled anti-PtdSer mAb are used to monitor apoptotic cells in vivo (please see Rybczynska et al, Med. Res. Rev. 38, 1713, 2018, for review). I feel from the data provided in the manuscript that the affinity of Apo-15 to PtdSer is much weaker and less specific than these compounds.

Reviewer #2 (Remarks to the Author):

This manuscript by Barth et al. describes fluorogenic peptide-based probes for in vivo imaging and quantitative evaluation of drug-induced apoptosis. The authors designed amphiphilic cyclic peptides with a fluorogenic Trp-BODIPY dye for targeting flipped phosphatidyl serines on outer membrane of apoptotic cells. They evaluated the fluorescence properties against apoptotic and non-apoptotic cells and retention in apoptotic cells. One of the probes, Apo-15, showed suitable fluorogenic property for selective staining of apoptotic cells. Using Apo-15, the authors performed quantitative evaluation of apoptotic cells. In addition, they detected drug-induced apoptosis in mice by ex vivo and intravital imaging.

The presented data showed fluorogenic labeling of drug-induced early apoptotic cells, and the authors well mentioned the advantages of the fluorogenic probe (Apo-15) over existing fluorescent probes for detecting apoptotic cells. Overall, this manuscript is acceptable for publication in Nature Communications. After addressing the following concerns, I will reconsider the manuscript for publication.

In the manuscript, the authors mentioned rational design of fluorogenic probe for imaging apoptotic cells. They mainly discuss the selection of cationic, aromatic, and hydrophobic amino acid residues to optimize fluorogenic peptide for labeling the target cells. However, the authors do not describe why they focus on cyclic peptides comprised of six residues as a scaffold of fluorogenic probe. The authors should explain about the reason for selection of this scaffold.

In Figure 1B, the authors quantified fluorescence labeling properties for screening of the probes. How many cell numbers for determination of the fluorescence signals? The detailed information on calculation should be described.

In Figure 2A, the apoptotic cell labeling efficiency seems different among cell types. In epithelial cells, the fluorescence signals of Apo-15 and annexin V are not clearly merged. The authors should

comment on the dependency and the quantitative data of labeling in different cell types should be shown if the authors have.

Although fluorogenic property is favorable, I think that fluorophore-labeled Annexin V can label apoptotic cells without a washing procedure because it binds to apoptotic acell membranes. Is the signal/background fluorescence ratio higher than that of Annexin V staining?

In imaging experiment in mice, the fluorescence signals were not found along membranes in the cells. In particular, under intravital condition, the snapshot images (Figure 5G) and the movie (Supplementary Movie 5) are not clear to reveal phagocytotic engulfment. Is it possible to show expanded images?

Reviewer #3 (Remarks to the Author):

Developing a technique to assess apoptosis in vivo, such as using intravital imaging, may allow us to address important questions that ex vivo techniques cannot. Here, the authors develop a peptide, Apo-15, which binds selectively to apoptotic (Annexin V+) cells independently of calcium concentration and with good retention after washes or protease/oxidant treatment. Apo-15 binding does not alter apoptosis or phagocytosis of apoptotic cells. In the LPS model of lung injury +/- CDKi, Apo-15 can be administered intratracheally and used to image apoptotic neutrophils using ex vivo flow cytometry or microscopy or intravital microscopy.

General Comments: This novel technique for labeling apoptotic cells in vivo could potentially be useful. However, several issues must be addressed with regards to the in vivo model. Importantly, while the labeling of BAL neutrophils is quite convincing, the labeling of neutrophils in vivo/in situ and the labeling of macrophages phagocytosing apoptotic neutrophils are not. Additionally, demonstration that Apo-15 can label apoptotic epithelial or endothelial cells, which are a critical component of lung injury, and/or apoptotic cells in another organ, would greatly strengthen the manuscript.

Specific Comments:

Major Comments:

1. There is concern that Apo-15 binding to apoptotic epithelial cells will not be as discriminatory between apoptotic and viable cells as binding to lymphoid and myeloid cells. In Fig. 2A, there is a cell

in the lower left corner of the field that is positive but dim for Annexin V and almost but not quite negative for Apo-15. The authors have not put an arrow on this cell, as it is likely unclear whether it is positive or negative. To raise confidence that the peptide will be effective for assessing epithelial cell apoptosis, a negative (unstained) control should be included in Fig. 2A to justify the exposure time and histogram settings chosen. Additionally, the percent stained cells should be quantified for epithelial cells, as done for myeloid and lymphoid cells in Fig. 2B. Also, Fig. 2B shows that Annexin V and Apo-15 both stain the same % of cells. It would be more convincing to show quantitatively that they stain the SAME cells (as is done for neutrophils in Fig. 2C). Assuming Annexin V is the gold standard, Fig. 2B should be expressed as the percent of Annexin V+ cells that are Apo-15+. Alternatively, lymphoid cells and epithelial cells should be shown in Fig. 3C.

2. Fig. 5D,E: Mice were administered LPS or LPS + CDKi, then i.t. Apo-15, and at 37-48 hours, lungs were fixed and imaged, and the number of Apo15+ cells per 0.25 μm^2 of lung was counted. These sections should be costained with Ly6G antibodies to confirm that the apoptotic cells identified in vivo are neutrophils, as there are many other cell types in the lung that could potentially be apoptotic during lung injury. Since the point of this figure is to show that Apo-15 stains all apoptotic cells and only apoptotic cells, these sections should also be costained with another marker of apoptosis (e.g., TUNEL) to confirm that these cells are apoptotic. Finally, a field showing a lung that did not get apo-15 should be shown as a negative control. A high power image w/ DAPI costaining should also be shown. These issues are particularly important since lung tissue (and particularly RBCs) tends to be autofluorescent in the green channel. Also, there is no data in fixed lung sections supporting the sentence on line 358 stating that "high-resolution tissue images revealed Apo-15-positive signals in phagocytic macrophages".

3. Suppl. Fig. 12: The authors conclude that BAL macrophages are Apo-15+, as shown by flow cytometry and cytopins. 48 hours is quite early in the LPS injury and resolution time course, occurring prior to the peak of neutrophil recruitment (PMID 21471090). Although there may be some uptake of apoptotic neutrophils at this early time point, it is a bit surprising that nearly half of alveolar macrophages would have phagocytosed apoptotic neutrophils. To justify the claim that these macrophages have phagocytosed apoptotic neutrophils, additional data is needed. First, the Apo-15 flow histogram from BAL macrophages from mice not administered Apo-15 should be shown to justify where the Apo15+ gate was drawn. Second, the authors reason that since CD45+ CD11c+ macrophages were not Apo15+ when stained with Apo15 ex-vivo, the macrophages found to be apo15+ after i.t. apo15 must have eaten apoptotic neutrophils. However, I cannot find the data showing that macrophages were not Apo15+ when stained with Apo15 ex-vivo. In addition, these macrophages should be stained ex vivo for annexin V to confirm that they are not apoptotic. Third, the cytopins appear to show mononuclear cells that are apo15+, but it would help if macrophage markers were also used. Additional staining for neutrophil markers would help support the claim that there are indeed neutrophils inside of these cells. Finally, and this is important, there do not appear to be any neutrophils on this cytopin (based on neutrophil nuclear morphology), which is inconsistent with the strong neutrophil predominance in BAL at this time point in the LPS model (Supplementary Fig. 10 and PMID 21471090).

4. Fig. 5F,G shows apo15+ cells in the lungs after LPS by intravital imaging. There are several issues that should be addressed here. (1) It is surprising that these images appear so similar, as one would expect that there would be movement of neutrophils around the lungs over a 1 hour period. (2) As

discussed above, there are several other cell types in the lungs that may be apoptotic. To demonstrate that these apo15+ cells are neutrophils, neutrophils should be labeled (e.g., with RFP), and there are several methods available to do this. Then, the % of neutrophils that are apoptotic as measured by intravital imaging should be determined. If Apo-15 is going to be used for intravital imaging, it is important to know what cell type it is staining in vivo. (3) The phagocytosis of apoptotic neutrophils by macrophages in the red dotted circle is not convincing. Labeling macrophages and neutrophils would be helpful. Improving this figure is critical, as the authors argue that the real impact of this peptide development is to allow intravital imaging.

5. The impact of this novel technique would be much higher if it could be demonstrated that cells other than neutrophils can be efficiently labeled. As discussed above, costaining of sections in Fig. 5D for epithelial/endothelial markers would demonstrate that Apo-15 can label other cells in vivo. Apo-15 administered intratracheally should at least be able to stain apoptotic alveolar type 1 and type 2 cells. Using transgenic mice with fluorescently labeled epithelial cells would allow detection of apoptotic epithelial cells using Apo-15 and intravital imaging. It is possible that a longer time course or a higher dose of LPS or even a different model of lung injury (e.g., bleomycin or influenza) will be necessary to induce significant epithelial cell apoptosis to test the apo-15 labeling, but this would greatly strengthen the manuscript. (This is also a concern since the ex vivo staining of epithelial cells in Fig. 2A is not convincing, as discussed above.) Labeling of apoptotic endothelial cells may require systemic administration of Apo-15 if the peptide cannot readily cross the injured alveolar barrier. Demonstrating that Apo-15 can be administered systemically will also be critical to label apoptotic cells in other organs. If Apo-15 can only be effectively administered i.t. and only really works to label hematopoietic cells, this will greatly limit its utility.

6. The authors state generally that being able to assess apoptosis in vivo, i.e., in intravital imaging, provides advantages over ex vivo staining. In addition to this general statement, it would be helpful to have concrete examples of gaps in our knowledge that are unable to be addressed by current techniques which could be studied w/ Apo-15.

Minor Comments:

1. Intratracheal administration always results in a patchy distribution throughout the lungs, so neutrophils in some airspaces will not be labeled. In the future, when Apo-15 is used to quantitatively compare neutrophil apoptosis under different conditions (e.g., wt vs. KO), this will be a confounding factor. This should be discussed. Systemic administration of Apo-15, if possible, would likely not have this limitation.

2. Suppl Fig. 11 is very strong data and should be moved to the main manuscript, Fig. 5C.

3. Fig. 4B should have unstained histogram for apo-15, similar to that shown for ANnexin V.

4. Fig. 5A: In lines 333-334, the authors state that r-roscovitine increases neutrophil apoptosis as determined by annexin V staining, but the annexin staining is not shown.

5. The word “unbiased” should be removed from line 353 because the methods are not truly unbiased.

Responses to reviewers

Reviewer 1

General comments:

1. The author's group previously published a similar cyclic peptide (called cLac-Bodipy) that was claimed specifically bind to negatively charged phospholipids including PtdSer. How apo-peptides are different from cLac-Bodipy?

Answer: We thank the reviewer for this important comment. The apo-peptides in this manuscript differ significantly from cLac-BODIPY, which is a longer (13-mer) fluorescent peptide designed to mimic the interactions between MFG-E8 (C2 lactadherin) and negatively-charged phospholipids. Our previous characterisation of the properties of cLac-BODIPY *in vitro* demonstrated binding to PtdSer and ability to label extracellular vesicles released by apoptotic BL-2 cells (reference 11 in the revised manuscript). However, even at high concentrations of cLac-BODIPY (in the μM range), we did not observe labeling of intact apoptotic cells. To illustrate the differences between cLac-BODIPY and Apo-15, we include Figure 1 for reviewers demonstrating that Apo-15 labels intact apoptotic BL-2 cells -which exhibit distinct FSC/SSC properties from apoptotic bodies- whereas cLac-BODIPY does not. We used AF647-Annexin V to confirm that light irradiation had successfully induced apoptosis in BL-2 cells. On the other hand, cLac-BODIPY did label a distinct population of subcellular material (on the basis of altered laser scatter properties) consistent with labeling of apoptotic bodies. As shown below, this subcellular material was also labeled with AF647-Annexin V and Apo-15, consistent with the data presented in our manuscript. We have included the features of cLac-BODIPY in revised Figure 1B and also included it in the text of the revised manuscript.

Figure 1 for reviewers. Flow cytometric analysis of BL-2 cells after induction of apoptosis and staining with Apo-15, cLac-BODIPY and AF647-Annexin V. Intact apoptotic cells are highlighted in blue.

2. *Biochemical characterization of apo-peptides is not sufficient. More detailed binding specificity to the phospholipids should be shown at least for Apo-15.*

Answer: This is an important point and we are thankful for the insight. In the revised manuscript, we have characterized the binding specificity of Apo-15 for different phospholipids. Specifically, we compared the binding of Apo-15 to PtdSer, cardiolipin, phosphatidylcholine, phosphatidylglycine, phosphatidylethanolamine and also phosphatidic acid. Given the fluorogenic nature of Trp-BODIPY, we used the fluorescence intensity of Apo-15 to measure its relative specificity. In these experiments, we observed that Apo-15 shows preferential binding to PtdSer in comparison to other phospholipids and to phosphatidic acid. Of note, Apo-15 showed significantly stronger binding to PtdSer than to phosphatidylethanolamine, which is also known to localize on the outer leaflet of the plasma membrane following induction of apoptosis (reference 25 in the manuscript). This new data has been included in the revised Figure 3.

3. *Several other groups designed PtdSer-binding peptides, but due to the insufficient affinity and specificity, none of them are successful for imaging apoptotic cells in vivo (for review, see please see Rycbczynska et al, Med. Res. Rev. 38, 1713, 2018.). Do the authors think that Apo-15 is better than others? Please discuss with more data (specificity and affinity).*

Answer: We thank the reviewer for the comment and suggestion. In the revised manuscript, we have included the suggested review as well as references on other PtdSer-binding peptides to in the Discussion of the revised manuscript. Our new experiments indicate that Apo-15 binds preferentially to PtdSer over other phospholipids (revised Figure 3). Importantly, we also present new data to show that Apo-15 can partially displace Annexin V binding to apoptotic cells (revised Figure 3 and Supplementary Figure 12). Unlike other reported PtdSer-binding peptides, a unique feature of Apo-15 is the inclusion of Trp-BODIPY as a reporter that emits increased fluorescence after binding in order to achieve high signal-to-noise ratios (in the range of 100-fold, as presented in the new Supplementary Figure 10 and discussed in the response to reviewer 2) with minimal background fluorescence for imaging. Altogether, Apo-15 represents an important addition to the toolbox of apoptosis reagents for fluorescence-based characterisation studies with calcium independence and real-time imaging capabilities with subcellular resolution.

Detailed comments:

1. *Figure 1B. Please include cLac-Bodipy in the Table for comparison. Please describe how the sequence of the 15 apo-peptides is designed.*

Answer: We have included cLac-Bodipy in the Table of Figure 1B.

A similar comment on the chemical design was made by reviewer 2, and therefore we have elaborated on the rationale and chemical features of apo-peptides: 1) cyclic core for high stability, 2) binding tunability by altering amino acids, 3) compatibility with Trp-BODIPY, 4) small size for easy manufacture and tissue accessibility. This information is now included in the revised manuscript (page 4). The manuscript also includes a stepwise description of the rational approach we employed in the design of the sequences of apo-peptides (pages 4 and 5). As discussed in the revised text, these steps included: 1) evaluation of amphipathic peptides with negatively- and positively-charged amino acids (Apo-0, Apo-2), 2) comparison of Trp and Phe as aromatic amino acids (Apo-2 to Apo-4), 3) intercalation of Trp residues within the amphipathic sequences (Apo-5 to Apo-8), 4) variable ratios of charged vs hydrophobic amino acids (Apo-11 to Apo-14). And finally, the optimisation of the clogP and the polar surface area of Apo-8 to obtain Apo-15 as the lead probe for optical detection of apoptotic cells with minimal staining of viable cells.

2. *Figure 1C: The idea of Bodipy was explained in the authors' own previous publications (reference 10-11). This Figure may not be necessary.*

Answer: We have modified Figure 1C to clarify the novel aspects of the work presented here. Although the Trp-BODIPY structure has been shown in other publications, the revised Figure 1C summarises the general structure of apo-peptides, their schematic behaviour in proximity to membranes of viable and apoptotic cells and the fluorescent pictograms of Apo-15, as elements that are unique to this manuscript.

3. *In the cell death field, APO-1 is known as a cell death receptor (Fas) (Cell Death Differ. 10, 26, 2013).*

Answer: We apologise for this oversight and have renamed “Apo-1” as “Apo-0”.

4. *Figure 2A: The authors claim that binding of Annexin V and Apo-15 to PtdSer is not competitive. What kinds of mechanism the authors propose? Annexin V and Apo-15 bind to different parts of PtdSer? In the middle panel of Figure 2, I can find a cell (at left) that is strongly bound by Annexin V, but only weakly bound by Apo-15, while the cell at middle has an opposite relation. I want to see the effect of high concentration of Annexin V on the Apo-15 binding.*

Answer: We thank the reviewer for this important insight and suggestion. We have performed competition assays between Annexin V and Apo-15 at different concentrations in co-cultures of viable and apoptotic cells. We observed that high concentrations of Apo-15 can reduce Annexin V binding to apoptotic cells and also that Annexin V reduces Apo-15 binding. These experiments suggest that Annexin V and Apo-15 compete for binding sites on the surface of apoptotic cells. However, we have found that the binding of Apo-15 and Annexin V cannot be fully blocked even at high concentrations of competing reagent, possibly due to the high levels of PtdSer that are present on apoptotic cell membranes. Our data also suggest that the affinity of Annexin V for apoptotic cells is likely to be higher than that of Apo-15. We believe that these are important new results and we have included them in the text, in the revised Figure 3 and in the new Supplementary Figure 12.

5. *Figure 2A legend: Why the concentration of Apo-15 (100 nM) and Annexin (0.01 µg/ml) are described in different ways (molar or mass concentration)? The molar concentration of 0.01 µg/ml Annexin is about 0.3 nM, indicating that the affinity of Apo-15 to apoptotic cells is 100 times weaker than that of Annexin. Yes?*

Answer: We thank the reviewer for the comment. We have carefully checked the concentrations of fluorescently-labeled Annexin V and realised that we had previously reported the wrong information as concentrations vary with different batches. The concentrations that we have used for AF647-Annexin V are now clearly indicated in the Figure legends, and they are in the low nanomolar range (e.g., 5-25 nM) in line with the concentrations reported by other groups (Martin et al., J. Exp. Med. 1995, 1545). The competition assays between Apo-15 and AF647-Annexin V (revised Figure 3) also indicate that Annexin V qualitatively shows stronger binding to apoptotic cells than Apo-15. However, it is difficult to quantitatively compare their fluorescence readouts because they rely on two different fluorophores with non-comparable optical properties (i.e., AlexaFluor647 for Annexin V and Trp-BODIPY for Apo-15). Our experimental data also emphasise the value of environmentally-sensitive reporters for imaging. Thus, despite qualitative differences in affinity, fluorescence cell images with similar signal-to-noise ratios (new Supplementary Figure 14) can be obtained using Annexin V or Apo-15 (examples of comparable images shown in Figures 1E, 2A and 3E).

6. *Figure 2C: If Annexin V and Apo-15 similarly bind to PtdSer on apoptotic cells, why the binding of Apo-15 to apoptotic cells is not proportional to that of Annexin V?*

Answer: The reviewer raises an important point related to proportional binding. We present data in the revised Figures 3 and S12 that demonstrate that Annexin V and Apo-15 are able to compete for binding on the surface of apoptotic cells. In our previous studies of PtdSer-binding opsonins, we observed an almost exact proportional binding to apoptotic cells between proteins containing the Gla domain (i.e., Protein S and Gas6) but not between these same proteins and Annexin V (Dransfield et al., *Cell Death Disease*, 2015, 6, e1646), suggesting that distinct molecular structures may bind PtdSer differently. As suggested in the literature (van Genderen et al., *Biochim Biophys Acta*. 2008, 953), Annexin V may assemble into trimers and form 2D lattice structures on the apoptotic cell surface. Whereas this is not fully known, we speculate that the multimerization of Annexin V may affect availability of sites for other PtdSer-binding proteins (protein S, Gas 6) or peptides (Apo-15).

7. *Figure 3B: I want to know whether Apo-15 binds to other phospholipids (cardiolipin, phosphatidylglycine, phosphatidylethanolamine, phosphatidylinositol, and phosphatic acids) or not. How is Kd (Dissociation Constant) of Apo-15 to PtdSer?*

Answer: As described above, we have analysed the binding of Apo-15 to PtdSer, cardiolipin, phosphatidylcholine, phosphatidylglycine, phosphatidylethanolamine and also phosphatidic acid. In these experiments, we observed that Apo-15 shows preferential binding to PtdSer in comparison to other phospholipids and to phosphatidic acid, and this new data has been included in the revised Figure 3. We believe that the accurate determination of KD values would require further biochemical experiments (e.g. superficial tension assays or surface plasmon resonance) in small unilamellar vesicles that are beyond the scope of this manuscript. However, we performed qualitative fluorescence titration assays where we measured the response of a fixed concentration of Apo-15 to increasing amounts of PtdSer. In these assays, we observed a clear concentration-dependent binding of Apo-15 to PtdSer and these results have been added to the revised manuscript (new Supplementary Figure 10).

8. *Figure 3C: Scramblases that are activated during apoptosis are non-specific for phospholipids. Thus, apoptotic cells expose not only PtdSer, but also other phospholipids such as phosphatidylethanolamine and phosphatidylinositol. Since apoptotic scramblases are not expressed in PLB-985 cells, not only PtdSer but also many other phospholipids (mentioned above) will not be exposed.*

Answer: We thank the reviewer for this point. Our experiments in PLB-985 cells confirm that externalisation of lipids via Xkr8 scramblase is a requirement for Apo-15 binding to apoptotic cells. We agree with the reviewer that other phospholipids (such as phosphatidylethanolamine) may not be exposed on the surface of PLB cells. As discussed above, our new data relating to the binding of Apo-15 to phospholipids (revised Figure 3) suggests that PtdSer is the main phospholipid being recognised by Apo-15. In light of the reviewer's comments, we have considered this point and address it in the revised manuscript.

9. *Figure 3C and 3D: According to Fig. 3D, about 20% of staurosporine-treated PLB are apoptotic. In Figure 3C, there is a population that is recognized by APO-15 but not by Annexin V. What is this population? They are not apoptotic? In Reference 7, binding of MFG-E8 to apoptotic cells was shown by double staining with MFG-E8 and TUNEL (DNA fragmentation). Is a similar approach not possible?*

Answer: We thank the reviewer for these observations. The cell populations the reviewer refers to represent a very small percentage (<5%) of the total cell population and appear prominent as a result of the way data is presented in the flow cytometry plots (i.e., single

events shown as dots). In the revised manuscript, we have included the relative percentages of Apo15+/Annexin V- cells present (bottom right quadrants) in Figure 3D to clarify this point.

10. *Figure 4B: It is well known that viable cells expose PtdSer under some circumstances. I am not sure what the authors want to say in this Figure. Do the authors want to say that Annexin V is much more sensitive than Apo-15 to detect PtdSer, or Annexin V binds non-specifically to viable cells? Please show the binding profiles of Annexin V and Apo-15 to apoptotic monocytes. Do they bind to apoptotic monocytes to the same extent?*

Answer: We agree with the reviewer that the difference in Annexin V and Apo-15 binding to viable monocytes may reflect their binding affinity for PtdSer (as revealed with our new data in the Annexin V vs Apo-15 competitive assays). We therefore consider that this figure does not significantly add to the main focus of this manuscript and we have excluded it from the revised manuscript.

11. *Figure 4C and D: Please describe how much Apo-15 and Annexin V were added to the phagocytosis assay.*

Answer: We initially used Annexin V at 25 nM and Apo-15 at 60 nM. In the revised manuscript, we have also included new experiments where we have used 10-fold higher concentration of Apo-15 (i.e., 600 nM) (revised Figure 4). We have added these experimental details in the revised manuscript.

12. *Figure 4: The authors claim that Annexin V inhibits the engulfment of apoptotic cells, but Apo-15 does not. This is misleading. A large amount of PtdSer is exposed on the surface of apoptotic cells. If most of the PtdSer is masked by Annexin V, the engulfment will be inhibited. Thus, the inhibition by Annexin V is dose-dependent. Similar dose-dependent inhibition of phagocytosis was shown with other PtdSer-binding protein (Reference 7). The concentration of Apo-15 used to inhibit the engulfment is the concentration at which the binding of Apo-15 to PtdSer is saturated? Do the authors want to say that the binding site of Apo-1 on PtdSer is different from that recognized by macrophage (or Protein S) for engulfment? Otherwise, it seems that Apo-15 recognizes a molecule rather than PtdSer so that PtdSer can work as an "eat me" signal.*

Answer: We agree with the reviewer that of the levels of PtdSer exposed on the surface of apoptotic cells are very high and that inhibition of apoptotic cell phagocytosis by Annexin V is concentration-dependent. Since Annexin V and Apo-15 show different working concentrations, we have tested their effects on phagocytosis in cultures where Annexin V or Apo-15 were present at *in vitro* working concentrations or at significantly higher concentrations. In phagocytosis assays using 10-fold higher concentrations of Apo-15 (up to 600 nM) we observed no significant differences in the phagocytic index (new data included in revised Figure 4). These data suggest that Apo-15 is unlikely to exert inhibitory effects on phagocyte clearance of apoptotic cells that would alter progression and ultimately resolution of inflammation.

13. *Page 3, line 55: Reference 7 is not a correct reference here.*

Answer: We have amended this in the revised manuscript (page 3).

14. *The authors compare Apo-15 with Annexin V, and claim that different from Annexin V, it does not require Ca^{2+} . It should be noted that Lactadherin (MFG-E8) C2 does not require Ca^{2+} , too.*

Answer: We have amended this in the revised manuscript.

15. Figure 5: The authors claim that Apo-15 can be used to detect apoptotic cells *in vivo*. But, technetium-labelled C2 domain of lactoadherin or synaptotagmin and I124 labeled anti-PtdSer mAb are used to monitor apoptotic cells *in vivo* (please see Rybczynska et al, *Med. Res. Rev.* 38, 1713, 2018, for review). I feel from the data provided in the manuscript that the affinity of Apo-15 to PtdSer is much weaker and less specific than these compounds.

Answer: The review article describing these reagents is relevant and we apologise for having overlooked it. We have now included it in the revised manuscript (reference 39). The reviewer highlights protein-based reagents that have been developed to image apoptosis using non-optical imaging, such as SPECT (single-photon emission computed tomography) or PET (positron emission tomography). However, none of these two agents can be used for optical imaging, and fluorescently-labeled annexins present some limitations, as highlighted in the manuscript. The need for calcium-insensitive and functionally-neutral optical reagents for apoptosis is the main driver for the development of Apo-15. Our new results have extended the characterisation of Apo-15 and clarified important aspects about its binding specificity. Altogether, the revised manuscript describes in full the chemical design and synthesis of Apo-15 as a completely new probe for imaging apoptosis with extensive *in vitro* characterisation and also proof-of-concept data showing its *in vivo* potential for monitoring drug-induced apoptosis in two different preclinical models. We believe that the experimental data in the revised manuscript strongly support the value of Apo-15 as an excellent optical probe to label apoptotic cells *in vitro* and *in vivo*.

Reviewer 2

In the manuscript, the authors mentioned rational design of fluorogenic probe for imaging apoptotic cells. They mainly discuss the selection of cationic, aromatic, and hydrophobic amino acid residues to optimize fluorogenic peptide for labeling the target cells. However, the authors do not describe why they focus on cyclic peptides comprised of six residues as a scaffold of fluorogenic probe. The authors should explain about the reason for selection of this scaffold.

Answer: We thank the reviewer for the comment. In the search for new chemical structures that could be used to image apoptotic cells *in vivo*, we decided to focus on small cyclic peptides because they offer several important advantages: 1) resistance to proteolytic cleavage and oxidative conditions for *in vivo* studies in inflammatory environments, 2) tunability of the chemical properties by changing the amino acid sequence, which allowed us to optimise binding to apoptotic cells, 3) compatibility with the fluorogenic Trp-BODIPY amino acid for wash-free imaging, and 4) smaller size than proteins (e.g., annexins) for improved ease of manufacture and tissue accessibility. These features are highlighted in the summary Table 1, and we have included this detailed rationale in the revised version of the manuscript.

In Figure 1B, the authors quantified fluorescence labeling properties for screening of the probes. How many cell numbers for determination of the fluorescence signals? The detailed information on calculation should be described.

Answer: These details were omitted for brevity, and we have now included them in the Online Methods section.

In Figure 2A, the apoptotic cell labeling efficiency seems different among cell types. In epithelial cells, the fluorescence signals of Apo-15 and annexin V are not clearly merged. The authors should comment on the dependency and the quantitative data of labeling in different cell types should be shown if the authors have.

Answer: We thank the reviewer for the comment. In experimental terms, the different cell populations that we have examined are heterogeneous, undergoing apoptosis to different extents and through different mechanisms (e.g., constitutive, UV-induced or chemically-induced apoptosis). Although it is difficult to quantify the relative labeling efficiency of Apo-15, we have used flow cytometry to demonstrate the differences in Apo-15 staining between viable and apoptotic cells in ALL lineages (i.e., myeloid, lymphoid and epithelial) and consistently in a calcium-independent manner (revised Figure 2).

In view of the reviewers' comments related to our data using the epithelial cell line A549, we have now examined Apo-15 labeling in primary airway epithelial cells in which apoptosis has been induced by treatment with staurosporine. We have found that the use of primary airway epithelial cells results in a more homogeneous response and that these conditions induce apoptosis with minimal necrosis. Furthermore, we observed good co-localization between the staining of Apo-15 and Annexin V-AF647, suggesting that the use of the A549 cell line contributed to the heterogeneity of the observed response. We have also quantified by flow cytometry the labeling efficacy of Apo-15 for apoptotic epithelial cells in media with or without calcium. The new fluorescence microscopy images and the new quantitative analysis have been added to the revised Figure 2.

Although fluorogenic property is favorable, I think that fluorophore-labeled Annexin V can label apoptotic cells without a washing procedure because it binds to apoptotic cell membranes. Is the signal/background fluorescence ratio higher than that of Annexin V staining?

Answer: We agree that the high binding affinity of Annexin V for apoptotic cells leads to good signal-to-noise ratios even when the protein is conjugated to always-on fluorophores (i.e.,

FITC, AlexaFluor647). To compare their signal-to-noise ratios, we have taken fluorescence images of apoptotic cells after staining with Apo-15 and Annexin-FITC under the same experimental settings (excitation: 488 nm). After image analysis, we observed similar signal-to-noise ratios for both (i.e., in the range of 100-fold, new Supplementary Figure 14). This result highlights the fact that Apo-15 can distinguish apoptotic cells to the same extent as annexins thanks to the use of an environmentally-sensitive fluorogenic reporter. Furthermore, as shown in Figure 2, Apo-15 performs to the same level in media with or without calcium, which represents a significant advantage over annexin-based reagents.

In imaging experiment in mice, the fluorescence signals were not found along membranes in the cells. In particular, under intravital condition, the snapshot images (Figure 5G) and the movie (Supplementary Movie 5) are not clear to reveal phagocytotic engulfment. Is it possible to show expanded images?

Answer: We thank the reviewer for the comment. We agree that it is difficult to unequivocally assign the phagocytic nature of Apo-15-stained cells *in vivo* using only intravital imaging, and we have clarified this point in the revised manuscript. As we also point out in our response to reviewer 3, we have included additional data to support our conclusion that Apo-15-labeled macrophages from mouse tissues can be the result of phagocytosis of apoptotic neutrophils. First, we have included flow cytometric data from BALs of LPS-treated mice that were not given Apo-15, which confirmed the lack of green fluorescence in viable CD45⁺CD11c⁺ Annexin V⁻ macrophages (new Supplementary Figure 15 and new Supplementary Figure 18 for the results). Second, as shown in the new Supplementary Figure 18, we have compared the histograms of viable macrophages from BALs of LPS-treated mice which had received Apo-15 *in vivo* or Apo-15 *ex vivo*. The brighter fluorescent signal of the former suggests that the acquisition of fluorescence by BAL macrophages is due to the *in vivo* phagocytosis of apoptotic cells. These results are also supported by our *in vitro* observations that Apo-15 does not impair the capacity of human MDMs to engulf dead cells (see response to reviewer 1 and revised Figure 4).

Furthermore, as shown in the Figure 2 for reviewers, we have performed new assays where human MDMs were incubated with apoptotic neutrophils, and we observed that Apo-15 only labeled phagocytic MDMs if Apo-15 was present when phagocytosis occurred (i.e., during incubation with dead cells) but not when phagocytosis had already occurred (i.e., after incubation with dead cells). In view of these results, we have clarified in the revised manuscript that Apo-15 may allow the identification of phagocytic cell populations.

Figure 2 for reviewers. Phagocytosis assays of human MDMs after incubation with apoptotic neutrophils and with incubation of Apo-15 during the assay (during phagocytosis) or after the assay (after phagocytosis).

Reviewer 3

Note from the authors in response to the comments from reviewer 3:

We thank the reviewer for the thorough review and many constructive suggestions. The revised manuscript includes new data where we have extended the *in vitro* characterization of Apo-15 and investigated its potential for labeling apoptotic cells *in vivo* in another mouse model of drug-induced apoptosis. Importantly, we have included new evidence that Apo-15 binds to PtdSer on the apoptotic cell membrane with selectivity over other lipids and that it competes for binding with Annexin V (revised Figure 3). We also present new data showing that Apo-15 labels primary airway epithelial cells that are apoptotic using fluorescence microscopy and flow cytometry (revised Figure 2). Finally, we have included new data that highlights the potential of Apo-15 in a different *in vivo* setting. Specifically, we employed a breast cancer model in which we induced apoptosis by treatment with cisplatin and administered Apo-15 intravenously *in vivo*; tumours were imaged and we compared the signals of Apo-15 and active caspase-3 (revised Figure 5). The results of these new experiments demonstrate that: 1) Apo-15 can be administered *in vivo* through different routes, and 2) Apo-15 labels cells *in vivo* that are also positive for other well-defined markers of apoptosis (e.g., active caspase-3).

The available evidence that we have to date is consistent with exposure of PtdSer being required for Apo-15 binding to cells. Given that Apo-15 labels apoptotic cells of different lineages -as shown in Figure 2 in the revised manuscript-, it is unlikely that Apo-15 labels apoptotic cells of a single origin *in vivo*. We suggest that undertaking extensive experimentation to unequivocally demonstrate this point would not add significantly to the detailed characterization of Apo-15 described in our manuscript. Indeed, universal and specific labeling of apoptotic cells with fluorescently-labeled Annexin V remains to be demonstrated *in vivo*. We have focused our revision to the use of Apo-15 in an *in vivo* setting, performing new experiments to address the following questions: 1) can Apo-15 be administered *in vivo* by different routes?, 2) do Apo-15 labeled cells co-localise with other markers of apoptosis (e.g., active caspase-3)?, and 3) does Apo-15 label cells in a different model of drug-induced apoptosis?. In addition, we have included additional important information requested by the reviewer (e.g., control experiments), which we did not include in the first version of the manuscript.

Specific comments:

Major comments:

1. There is concern that Apo-15 binding to apoptotic epithelial cells will not be as discriminatory between apoptotic and viable cells as binding to lymphoid and myeloid cells. In Fig. 2A, there is a cell in the lower left corner of the field that is positive but dim for Annexin V and almost but not quite negative for Apo-15. The authors have not put an arrow on this cell, as it is likely unclear whether it is positive or negative. To raise confidence that the peptide will be effective for assessing epithelial cell apoptosis, a negative (unstained) control should be included in Fig. 2A to justify the exposure time and histogram settings chosen. Additionally, the percent stained cells should be quantified for epithelial cells, as done for myeloid and lymphoid cells in Fig. 2B. Also, Fig. 2B shows that Annexin V and Apo-15 both stain the same % of cells. It would be more convincing to show quantitatively that they stain the SAME cells (as is done for neutrophils in Fig. 2C). Assuming Annexin V is the gold standard, Fig. 2B should be expressed as the percent of Annexin V+ cells that are Apo-15+. Alternatively, lymphoid cells and epithelial cells should be shown in Fig. 3C.

Answer: We thank the reviewer for the comment. In the revised Figure 2, we present fluorescence microscopy images of Apo-15 and AF647-Annexin V labeling of neutrophils, BL-

2 cells and new data for primary airway epithelial cells -in replacement of A549 cells-, demonstrating specific labeling of apoptotic cells from different lineages. Representative images were chosen because they show positive and negatively stained cells most clearly, thereby serving to illustrate the lack of staining of viable cells. However, demonstration that Apo-15 and Annexin V label the same cells is best illustrated by our flow cytometric data. In the new Figure 2B, we included representative flow cytometry histograms for neutrophils and BL-2 cells. The flow cytometry data for staurosporine-treated primary epithelial cells showing double positive Apo-15 and Annexin V apoptotic cells is shown below (Figure 3 for reviewers) for clarification. In addition, we also provide quantification of the percentages of Apo-15 and Annexin V-labeled cells for all three lineages including primary epithelial cells (new Figure 2D). Altogether, these results demonstrate that Apo-15 effectively discriminates between viable and apoptotic cells of different origins.

Figure 3 for reviewers. Flow cytometry analysis of staurosporine-treated primary airway epithelial cells after staining with Apo-15 and AF647-Annexin V.

We thank the reviewer for the comment on replotting the data in Fig 2B. However, we think that it would not be possible to present the data as the reviewer suggests because Annexin V does not bind to apoptotic cells under Ca^{2+} -free conditions (i.e., in the presence of EDTA). Therefore, we have left the data in Figure 2D as the staining for Apo-15 and Annexin V separately. The representative dot plots shown in Figure 2B and Figure 3 for the reviewers for epithelial cells show that most Apo-15-labeled cells also co-stain with Annexin V.

2. Fig. 5D,E: Mice were administered LPS or LPS + CDKi, then i.t. Apo-15, and at 37-48 hours, lungs were fixed and imaged, and the number of Apo15+ cells per 0.25 μm^2 of lung was counted. These sections should be costained with Ly6G antibodies to confirm that the apoptotic cells identified in vivo are neutrophils, as there are many other cell types in the lung that could potentially be apoptotic during lung injury. Since the point of this figure is to show that Apo-15 stains all apoptotic cells and only apoptotic cells, these sections should also be costained with another marker of apoptosis (e.g., TUNEL) to confirm that these cells are apoptotic. Finally, a field showing a lung that did not get apo-15 should be shown as a negative control. A high power image w/ DAPI costaining should also be shown. These issues are particularly important since lung tissue (and particularly RBCs) tends to be autofluorescent in the green channel. Also, there is no data in fixed lung sections supporting the sentence on line 358 stating that "high-resolution tissue images revealed Apo-15-positive signals in phagocytic macrophages".

Answer: Whilst we were able to identify Apo-15 positive apoptotic neutrophils in BALF using flow cytometry, we did not claim that Apo-15-labeled cells present in lung tissue sections were exclusively neutrophils. We agree with the reviewer that there may be other apoptotic cells that are present in the lung which could be labeled with Apo-15. Despite using multiple

conditions to undertake the experiments that the reviewer suggested, we have been unable to co-stain lung tissue sections with Apo-15 and Ly6G antibodies or other potential neutrophil markers (e.g., calprotectin) in order to identify Apo 15-positive cells as being neutrophils. The processing and antigen retrieval methods that we used (e.g., de-paraffinization and pressure cooking at 105°C in sodium citrate buffer) resulted in marked loss of the Apo-15 signal, which makes any co-staining with antibody-based markers very difficult. However, the flow cytometry data presented in the revised Figure 5 (former Supplementary Figure 11) confirms that a proportion of neutrophils (i.e., 5% in LPS-treated and 29% in LPS+CDKi-treated) in BAL samples are labeled with Apo-15.

Given the difficulties of preserving tissue architecture when working with frozen material obtained from the lung, we sought an alternative animal model of *in vivo* drug-induced apoptosis. We used a mouse model of breast cancer where we induced apoptosis by treatment with cisplatin. In this model, we were able to retain Apo-15 staining in tumour tissues (OCT embedded and frozen sectioned) and also label for active caspase-3 (new Figure 5). This new data demonstrates that Apo-15 can be administered intravenously *in vivo* and also that Apo-15 can label apoptotic cells as defined by activity of caspase 3.

We have now also included data of negative controls as requested by the reviewer, including lung tissue obtained from mice which were treated with LPS and CDKi but were not administered Apo-15 (new Supplementary Figure 16 and new Supplementary 17).

Finally, with regards to the comment on sentence 358, we agree with the reviewer and, in line with the suggestion, we have removed our comment suggesting that high resolution images reveal that Apo-15 signal can be detected in phagocytic macrophages and we further clarify this point in subsequent answers of our response.

3. Suppl. Fig. 12: The authors conclude that BAL macrophages are Apo-15+, as shown by flow cytometry and cytopins. 48 hours is quite early in the LPS injury and resolution time course, occurring prior to the peak of neutrophil recruitment (PMID 21471090). Although there may be some uptake of apoptotic neutrophils at this early time point, it is a bit surprising that nearly half of alveolar macrophages would have phagocytosed apoptotic neutrophils. To justify the claim that these macrophages have phagocytosed apoptotic neutrophils, additional data is needed. First, the Apo-15 flow histogram from BAL macrophages from mice not administered Apo-15 should be shown to justify where the Apo15+ gate was drawn. Second, the authors reason that since CD45+ CD11c+ macrophages were not Apo15+ when stained with Apo15 ex-vivo, the macrophages found to be apo15+ after i.t. apo15 must have eaten apoptotic neutrophils. However, I cannot find the data showing that macrophages were not Apo15+ when stained with Apo15 ex-vivo. In addition, these macrophages should be stained ex vivo for annexin V to confirm that they are not apoptotic. Third, the cytopins appear to show mononuclear cells that are apo15+, but it would help if macrophage markers were also used. Additional staining for neutrophil markers would help support the claim that there are indeed neutrophils inside of these cells. Finally, and this is important, there do not appear to be any neutrophils on this cytopin (based on neutrophil nuclear morphology), which is inconsistent with the strong neutrophil predominance in BAL at this time point in the LPS model (Supplementary Fig. 10 and PMID 21471090).

Answer: We agree with the reviewer that the time course of lung inflammation is critical for understanding the *in vivo* labeling data. In our experimental system, we have used a dose of 1 µg of LPS, which is around 1/20 of the low dose of LPS used in the elegant study of macrophage dynamics by Janssen *et al.* (Am. J. Respir. Crit. Care Med. 2011, 184, 547). In our previously published data using this dose of intratracheal LPS (Lucas *et al.*, Mucosal Immunol. 2014, 7, 857), we showed that the extent of neutrophil recruitment is lower than that observed by Janssen *et al.* and that neutrophil numbers are declining after 36 h, which are indicative of ongoing resolution of inflammation and consistent with the presence of significant

numbers of phagocytic macrophages. Therefore, we have revised the text to clarify the dose of LPS used in our animal experiments, both in the Methods and in the legend of Figure 5.

We agree with the reviewer that additional data would support our conclusion that Apo-15-labeled macrophages have phagocytosed apoptotic cells, and we apologise for not having included these data in the original submission. We have included our gating strategy (revised Supplementary Figure 15) and additional data to support our conclusion that green fluorescent macrophages present in BALs may arise as a consequence of phagocytosis of Apo-15 labeled apoptotic cells. First, we have included flow cytometric data from BALs of LPS-treated mice that were not given Apo-15, which confirmed the lack of green fluorescence in viable (i.e., Annexin V-negative) CD11c+ macrophages (gating in Supplementary Figure 15 and new data in Supplementary Figure 18). Second, we have compared the histograms of viable macrophages from BALs of LPS-treated mice which had received Apo-15 *in vivo* or Apo-15 *ex vivo*. The brighter fluorescent signal of the former suggests that the BAL macrophages acquire fluorescence due to the phagocytosis of apoptotic cells *in vivo* (Supplementary Figure 18). These results are also supported by our *in vitro* observations that even at high concentrations, Apo-15 does not impair the capacity of human MDMs to engulf dead cells (see response to reviewer 1 and revised Figure 4).

Furthermore, as pointed out in our response to reviewer 2, we have performed assays where human MDMs were incubated with apoptotic neutrophils and observed that Apo-15 only labeled phagocytic MDMs if Apo-15 was present when phagocytosis occurred (i.e., during incubation with dead cells) but not when phagocytosis had already occurred (i.e., after incubation with dead cells). This data is presented again in the Figure 2 for the reviewers below.

Figure 2 for reviewers (same as shown in response to reviewer 2). Phagocytosis assays of human MDMs after incubation with apoptotic neutrophils and with incubation of Apo-15 during the assay (during phagocytosis) or after the assay (after phagocytosis).

4. Fig. 5F,G shows apo15+ cells in the lungs after LPS by intravital imaging. There are several issues that should be addressed here. (1) It is surprising that these images appear so similar, as one would expect that there would be movement of neutrophils around the lungs over a 1 hour period. (2) As discussed above, there are several other cell types in the lungs that may be apoptotic. To demonstrate that these apo15+ cells are neutrophils, neutrophils should be labeled (e.g., with RFP), and there are several methods available to do this. Then, the % of neutrophils that are apoptotic as measured by intravital imaging should be determined. If Apo-15 is going to be used for intravital imaging, it is important to know what cell type it is staining in vivo. (3) The phagocytosis of apoptotic neutrophils by macrophages in the red dotted circle is not convincing. Labeling macrophages and neutrophils would be helpful. Improving this figure is critical, as the authors argue that the real impact of this peptide development is to allow intravital imaging.

Answer: We agree with the reviewer that there are limitations in the intravital imaging experiments which could make the data difficult to interpret. Furthermore, the lungs in our intravital experiments are imaged without perfusion so they may differ from lung tissues that were harvested for *ex vivo* analysis (i.e., tissue sections and BALs).

1) With respect to movement of Apo-15-labeled cells in the lung over a 1 h period, apoptotic cells would be expected to exhibit restricted motility, in part due to loss of cytoskeletal organisation and uncoupling of chemokine receptor signaling and integrin activity (Whyte *et al.*, *J Immunol.* 1993, 150, 5124–5134).

2) As discussed in the response to point 2, we did not claim that Apo-15 labeled cells visualised by intravital microscopy were exclusively neutrophils. Although the experimental strategy outlined by the reviewer would allow us to identify whether neutrophils were labeled with Apo-15, our *in vitro* data suggest it is likely that Apo-15 will also label other apoptotic cells *in vivo*. In our opinion, the definitive identification of the cellular reactivity of Apo-15 in every *in vivo* setting represents a significant undertaking that is beyond the scope of this manuscript. In this manuscript we describe in full the rational design, chemical synthesis and complete *in vitro* characterization of Apo-15 reactivity. Furthermore, our observations of Apo-15 labeling *ex vivo* and *in vivo* by intravital microscopy represent proof of concept data for the application of Apo-15 to study drug-induced apoptosis in dynamic *in vivo* processes and in different preclinical models.

3) As discussed above, we agree with the reviewer that microscopy is not a convincing way to define phagocytosis of apoptotic cells. We have therefore removed our comment suggesting that high resolution images reveal that Apo-15 signal can be detected in phagocytic macrophages.

We agree with the reviewer that intratracheal administration can lead to patchy distribution of staining throughout the lungs. In view of this, we now include lower magnification images of the lungs in the revised Figure 5 and have removed Supplementary Movies 4 and 5, which do not add significant new information (also in line with the comment of limited cell mobility above). The new images in Figure 5 emphasize the patchy distribution observed and are more representative of the Apo-15 staining that we have observed in the lungs of LPS-treated mice by intravital imaging. Quantification of the labeled cells after *in vivo* administration of Apo-15 is more readily achieved by our flow cytometry data, in which we could co-analyse the Apo-15 staining with antibodies to profile the different cell populations in the tissue. In the revised Figure 5 (former Supplementary Figure 11), we confirmed that a proportion of CD45+Ly6G+ neutrophils (i.e., 5% in LPS-treated and 29% in LPS+CDKi-treated) are labeled with Apo-15. The cell markers and gating strategies used for our flow cytometric analysis are illustrated in the Supplementary Figure 15.

We have also addressed the previous limitations of intratracheal administration by performing additional animal experiments where we administered Apo-15 intravenously. Specifically, we employed a breast cancer model in which we induced apoptosis by treatment with cisplatin and administered Apo-15 intravenously *in vivo*; tumours were imaged, and we compared the signals of Apo-15 and active caspase-3 (revised Figure 5). The results of these new experiments demonstrate that: 1) Apo-15 can be administered *in vivo* through different routes, and 2) Apo-15 label cells *in vivo* that are also stained with other well characterized markers of apoptosis (e.g., active caspase-3).

5. The impact of this novel technique would be much higher if it could be demonstrated that cells other than neutrophils can be efficiently labeled. As discussed above, costaining of sections in Fig. 5D for epithelial/endothelial markers would demonstrate that Apo-15 can label other cells in vivo. Apo-15 administered intratracheally should at least be able to stain apoptotic alveolar type 1 and type 2 cells. Using transgenic mice with fluorescently labeled epithelial cells would allow detection of apoptotic epithelial cells using Apo-15 and intravital imaging. It is possible that a longer time course or a higher dose of LPS or even a different model of lung injury (e.g., bleomycin or influenza) will be necessary to induce significant

epithelial cell apoptosis to test the apo-15 labeling, but this would greatly strengthen the manuscript. (This is also a concern since the ex vivo staining of epithelial cells in Fig. 2A is not convincing, as discussed above.) Labeling of apoptotic endothelial cells may require systemic administration of Apo-15 if the peptide cannot readily cross the injured alveolar barrier. Demonstrating that Apo-15 can be administered systemically will also be critical to label apoptotic cells in other organs. If Apo-15 can only be effectively administered i.t. and only really works to label hematopoietic cells, this will greatly limit its utility.

Answer: As mentioned above, we agree with the reviewer that there may be other apoptotic cells that are present in the lung which could be labeled with Apo-15. This is also supported by our results shown in Figure 2, where we corroborated that apoptotic cells of different origin -other than hematopoietic cells- can be effectively labeled by Apo-15. The reviewer points at whether Apo-15 can 'label other cells *in vivo*' and whether it can 'only be effectively administered *i.t.*'. These are two important points and we have addressed them by administering Apo-15 *in vivo* intravenously in a breast cancer mouse model where we induced apoptosis in tumours by treatment with cisplatin. We have included the new data in the revised Figure 5, which also demonstrates good co-localization between Apo-15 and active caspase-3 staining.

Finally, the reviewer suggests studies using animals with different specific cell lineage markers or different experimental models of injury in the lung, which would provide additional information related to the utility of Apo-15 for *in vivo* imaging in a wide variety of contexts and experimental situations. However, after carefully considering the reviewer's comments and the already available data, we suggest that the identification of the lineage of the cells that are labeled with Apo-15 when examined using intravital imaging *in vivo* represents an extensive series of additional studies that would require us to obtain mice with different lineage tracing reporters and to establish new models of injury, inflammation and repair. We suggest that this is beyond the scope of this present manuscript. In the revised manuscript, we have re-written the text to highlight the value and novelty of our work, which covers the chemical design and synthesis of Apo-15, extensive *in vitro* characterisation of Apo-15 labeling of apoptotic cells, and data showing the *in vivo* potential for use of Apo-15 to monitor drug-induced apoptosis. In the revised Discussion we now include consideration of the limitations of use of Apo-15, including the interesting possibilities raised by the reviewer.

6. The authors state generally that being able to assess apoptosis in vivo, i.e., in intravital imaging, provides advantages over ex vivo staining. In addition to this general statement, it would be helpful to have concrete examples of gaps in our knowledge that are unable to be addressed by current techniques which could be studied w/ Apo-15.

Answer: We thank the reviewer for the comment. In the revised manuscript, we have discussed potential future applications of Apo-15 as a new tool for imaging apoptosis, including some examples of inflammatory diseases.

Minor Comments:

1. Intratracheal administration always results in a patchy distribution throughout the lungs, so neutrophils in some airspaces will not be labeled. In the future, when Apo-15 is used to quantitatively compare neutrophil apoptosis under different conditions (e.g., wt vs. KO), this will be a confounding factor. This should be discussed. Systemic administration of Apo-15, if possible, would likely not have this limitation.

Answer: As the reviewer points out, intratracheal administration can lead to patchy distribution of staining throughout the lungs, so we now include lower magnification images in the revised Figure 5 in the revised manuscript to illustrate this point. Moreover, we have addressed the issue about the administration of Apo-15 *in vivo* by performing administration intravenously in

another mouse model (i.e., breast cancer mouse model) where we induced apoptosis by treatment with cisplatin. Moreover, using this approach we were able to demonstrate co-labeling of Apo-15 with the presence of active caspase 3. We have included this new data in the revised Figure 5.

2. *Suppl Fig. 11 is very strong data and should be moved to the main manuscript, Fig. 5C.*

Answer: We thank the reviewer for the comment. We have moved the data from the previous Supplementary Figure 11 to the main manuscript (revised Figure 5).

3. *Fig. 4B should have unstained histogram for apo-15, similar to that shown for Annexin V.*

Answer: The previous Figure 4B already included the unstained histogram for Apo-15; however, it wasn't easy to see as it overlapped with the Apo-15 stained sample. In any case, in view of the comments from reviewer 1 (see response to reviewer 1), we have replaced this with new data in the revised Figure 4.

4. *Fig. 5A: In lines 333-334, the authors state that r-roscovitine increases neutrophil apoptosis as determined by annexin V staining, but the annexin staining is not shown.*

Answer: We thank the reviewer for the comment. As it has been described in previous reports (Nat Med. 2006, 1056; FASEB J. 2009, 844; Eur J Immunol. 2010, 1127; Cell Death Differ. 2012, 1950), *R-roscovitine* can increase neutrophil apoptosis. We include a figure below (Figure 4 for reviewers) to clarify this point.

Figure 4 for reviewers. Flow cytometry analysis of neutrophil apoptosis. Representative histograms of forward scatter versus AF647-Annexin V staining. (left: untreated, centre: 2 h *R-roscovitine*, right: 3 h *R-roscovitine*) revealing induction of neutrophil apoptosis after 3h culture in the presence of *r-roscovitine*.

5. *The word “unbiased” should be removed from line 353 because the methods are not truly unbiased.*

Answer: We have removed the term unbiased.

Reviewers' comments:

Reviewer #1 (Remarks to the Author):

The authors revised the manuscript essentially according to my comments. I still have some comments. In particular, the discussion why the cyclic peptide (RKKWF) binds to PS but not PE (and others) is needed. My detailed comments are as follows.

Line 49: "DNA modification" is too vague. "DNA fragmentation" would be better.

Line 55: During apoptotic cell death, the scramlase (Xkr8) is activated. But, flippases (ATP11A and ATP11C) are inactivated. Please see Nagata et al. (Cell Death Diff. 23, 952-961, 2016).

Line 59: The PS-exposure is downstream of the caspase cascade (please see the above review by Nagata et al.), and theoretically later than the caspase activation. The PS-exposure should occur at the same time as the DNA fragmentation that is also caspase-dependent. Please remove "display faster kinetics and". A similar description in lines 436-438 should be corrected, too.

Line 91: Please how or why the first peptides (EEEWW and KKKWW) were designed.

Line 114: In several places, the authors say "constitutive apoptosis". What does it mean by "constitutive"? Does the incubation of neutrophils not induce their senescence?

Line 198: EDTA chelates not only Ca²⁺ but also Mg²⁺. EGTA is a more specific chelator for Ca²⁺. EGTA should be used here.

Line 222: Please explain more why necrotic and apoptotic cells give the different fluorescence lifetime. Figure 9 shows that necrotic cells carry the probe inside of cells, while the probe binds on the cell surface of apoptotic cells.

Line 247-248: It is interesting to see APO-15 distinguishes PS from PE. But please discuss a peptide (RKKWF) has the ability to distinguish them.

Line 269 and Supplementary Figure 13: Figure 13 does not show the DNA fragmentation. The data is very poor.

Line 304 and Figure 4A: How the apoptosis was assayed?

Line 364-365: Do the authors think that the formalin-treatment does not dissociate APO-15 from PS or it has no effect on APO-15-labelled apoptotic cells that had been engulfed by macrophages?

Reviewer #2 (Remarks to the Author):

In the manuscript, the authors described the detailed reasons for the use of the cyclic peptides scaffold, which highlights the rationality of the design of the fluorogenic probes. The authors also performed additional flow cytometric experiments to support the availability of the fluorogenic probes for apoptotic cell imaging in vivo. The presented data sufficiently addressed my concerns. Hence, I think that the revised manuscript would be acceptable for publication in Nature Communications without further modifications.

Reviewer #3 (Remarks to the Author):

The investigators have addressed many of the concerns and the manuscript is greatly improved. Concerns which remain:

1. The impact of a novel method lies in scientific questions that were unable to be addressed by previous methods and are now able to be addressed using the novel method. Therefore, to assess the impact of Apo-15, it is critical to know exactly what scientific questions will now be addressable that previously were not. In the context of lung injury, it is not convincing that Apo-15 will allow us to answer new questions. Most studies in fixed lung can already be performed using staining such as cleaved caspase 3 and TUNEL. The authors argue that live imaging is a distinct advantage of the new method but without being able to demonstrate what cell types are Apo-15+ in vivo and whether Apo-15 even labels cells other than neutrophils in vivo, it is difficult to imagine how this would be useful. This was the concern stated in comment 6 of the initial review. The authors did suggest at least one intriguing example, pulse dosing of Apo-15 to track the fate of apoptotic cells but they acknowledge that more work must be done to show feasibility. The authors argue that additional studies are beyond the scope of this manuscript, and indeed they have done quite a bit of work designing and synthesizing the compound, but without these additional feasibility studies, it is difficult to judge whether Apo-15 will ultimately be useful to answer important scientific questions.

2. At the bottom of p. 9, the authors state that "BALF and lung digests from LPS-only and LPS+CDKi mice were analyzed by multiparameter flow cytometry" but the digest data is difficult to find.

3. In response to concern shared by 2 reviewers that the images were unconvincing, the authors have removed images proposed to show engulfment of apoptotic neutrophils by macrophages. However, the remaining data supporting the claim that Apo-15 labels macrophages that have

ingested apoptotic neutrophils is not very strong: In Suppl. Fig. 18, the CD45+ CD11c+ Annexin- cells in the BAL are Apo-15+ when the Apo-15 is administered in vivo but these macrophages are not labeled by Apo-15 administered ex vivo. The authors argue that these data demonstrate that the macrophages have ingested apoptotic neutrophils. This may be the case but there may be other explanations for these data. Directly demonstrating that these cells have ingested neutrophils by including once again the cytopins but with staining for macrophage and neutrophil markers would be much stronger.

Response to reviewers' comments (NCOMMS-19-07315B)

Reviewer 1

The authors revised the manuscript essentially according to my comments. I still have some comments. In particular, the discussion why the cyclic peptide (RKKWF) binds to PS but not PE (and others) is needed.

Answer: We thank the reviewer for the comment which we have now addressed in the revised manuscript. Specifically, the screening of the 15 apo-peptides containing different amino acids -which included tryptophan, phenylalanine, leucine, valine and isoleucine as hydrophobic residues, and lysine, glutamic acid and arginine as polar residues- allowed us to establish some basic structure-activity relationships that define the chemical features for binding to the surface of apoptotic cells. Two key observations from this study were: 1) positively-charged residues (e.g., arginine, lysine) are crucial for effective binding, as shown by the lack of staining with the negatively-charged apo-peptide Apo-0; 2) positively-charged residues with large surface polar area (i.e., arginine) did improve signal-to-noise ratios and reduce off-target binding, as shown in the optimisation from Apo-3 to Apo-15. These two points suggest that the electrostatic interactions between apo-peptides and PtdSer are a major driving force for the binding.

The chemical structures of PtdSer and PE are similar. PtdSer and PE consist of the same hydrophobic chains, but they present differences in their headgroups. Specifically, PtdSer contains one free negatively-charged carboxylate group, which is not present in PE. We hypothesize that the stronger binding between Apo-15 and PtdSer may be due to the favourable electrostatic interactions between the free negatively-charged carboxylate group in PtdSer and the positively-charged residues in Apo-15. This explanation is also in agreement with the reduced binding of Apo-15 to PE, which does not have any free negatively-charged carboxylate group. These points are now considered in the revised Discussion of the manuscript (page 11).

Line 49: "DNA modification" is too vague. "DNA fragmentation" would be better.

Answer: We agree that DNA fragmentation is a more accurate description and thus we have amended the text in the revised manuscript accordingly.

Line 55: During apoptotic cell death, the scramblase (Xkr8) is activated. But, flippases (ATP11A and ATP11C) are inactivated. Please see Nagata et al. (Cell Death Diff. 23, 952-961, 2016).

Answer: We thank the reviewer for the comment. We have amended the text in the revised manuscript. The reference to Nagata et al. was already included as reference 7.

Line 59: The PS-exposure is downstream of the caspase cascade (please see the above review by Nagata et al.), and theoretically later than the caspase activation. The PS-exposure should occur at the same time as the DNA fragmentation that is also caspase-dependent. Please remove "display faster kinetics and". A similar description in lines 436-438 should be corrected, too.

Answer: The reviewer correctly points out the precise temporal sequence of events associated with apoptosis and we have amended the text accordingly in the revised

manuscript. We would like to point out that in our experience, the catastrophic loss of membrane asymmetry associated with apoptosis allows detection of PtdSer exposure prior to detection of extensive DNA fragmentation, potentially offering a more practical approach for the robust early detection of apoptotic cells.

Line 91: Please how or why the first peptides (EEEWW and KKKWW) were designed.

Answer: We designed apo-peptides as small cyclic peptides because they have several important advantages as imaging probes: 1) resistance to proteolytic cleavage for *in vivo* use, 2) tunability by changing the amino acid sequence, 3) compatibility with fluorogenic amino acids, and 4) smaller size than proteins for improved tissue accessibility.

Small amphipathic chemical structures have been previously reported for the detection of apoptotic cells (for instance, Aposense compounds as shown by Damianovich, M. et al. reference 48 in the manuscript). Therefore, we designed our first peptides (Apo-0: EEEWW, Apo-2: KKKWW) as two similar amphipathic peptides but with different net charges so that we could examine the importance of electrostatic interactions for binding to the surface of apoptotic cells. Apo-0 and Apo-2 contain the same number of hydrophobic amino acids (2 Trp and 1 Trp-BODIPY), have very similar molecular weight (Apo-0: 1324 Da, Apo-2: 1321 Da), and clog P values (Apo-0: -2.97, Apo-2: -2.78), but different net charges (Apo-0: -3 net charge, Apo-2: +3 net charge). With this design, we confirmed that positively-charged amphipathic peptides bind more strongly to apoptotic cells than negatively-charged amphipathic peptides. Therefore, our subsequent designs were focused only on peptides with positive net charges, and we generated apo-peptides with other chemical groups to examine the influence of aromatic vs non-aromatic hydrophobic residues (Apo 3-4 and Apo 9-10), alternate vs sequential positive charges (Apo 5-8) and also different polarities (Apo 11-14). These points are considered in the main text of the manuscript (page 4).

Line 114: In several places, the authors say "constitutive apoptosis". What does it mean by "constitutive"? Does the incubation of neutrophils not induce their senescence?

Answer: In the revised manuscript, we have changed the term 'constitutive apoptosis' to 'tissue culture-induced apoptosis' to avoid any confusion with neutrophil senescence. *In vitro* culture is a well-established technique for induction of neutrophil apoptosis (as shown in Savill et al., J. Clin. Invest. (1989), 83, 865-875 and in Dransfield et al., reference 55 in the manuscript) where neutrophils cultured *in vitro* for 18-24 hours undergo apoptosis, as defined by Annexin V binding, caspase activation and morphological changes (i.e., nuclear pyknosis and cytoplasmic vacuolation, DNA fragmentation, CD16 shedding) without the need for additional treatments (e.g., chemicals, UV light).

Line 198: EDTA chelates not only Ca²⁺ but also Mg²⁺. EGTA is a more specific chelator for Ca²⁺. EGTA should be used here.

Answer: We thank the reviewer for the comment. We have amended the text and clarified in the line 198 that EDTA acts to chelate divalent cations. We have also compared the capacity of Apo-15 to recognize apoptotic neutrophils in media containing Ca²⁺, EDTA and EGTA. As shown in the Figure for the reviewers 1 below, Apo-15 staining remains intact under all conditions whilst Annexin V does not bind upon chelation of divalent cations with EDTA or EGTA. This result strengthens the utility of Apo-15 over Annexin V for studies in low concentrations of divalent cations such as calcium and magnesium.

Figure 1 for reviewers. Flow cytometric analysis of mixtures of apoptotic and viable neutrophils after staining with AF647-Annexin V (25 nM) and Apo-15 (100 nM) in the presence of 2 mM CaCl₂, 2.5 mM EDTA or 2.5 mM EGTA. Representative histograms showing Apo-15 binding (x-axis, positive cells highlighted in green) vs AF647-Annexin V binding (y-axis) acquired on a 5L LSR flow cytometer.

Line 222: Please explain more why necrotic and apoptotic cells give the different fluorescence lifetime. Figure 9 shows that necrotic cells carry the probe inside of cells, while the probe binds on the cell surface of apoptotic cells.

Answer: We thank the reviewer for the comment. Trp-BODIPY is an environmentally-sensitive fluorophore and therefore it displays different fluorescence lifetimes depending on the chemical nature of the surrounding environment. As shown in Figure S9, we have observed that the fluorescence lifetime of Apo-15 is slightly different (around 1 ns longer) in apoptotic cells when compared to necrotic cells. Whereas deciphering the exact molecular mechanism behind this behavior would require extensive spectroscopic studies that are beyond the scope of this manuscript, we believe that Apo-15 is mainly binding to PtdSer in apoptotic cells while it could potentially bind to other molecules (e.g., phospholipids, proteins) inside necrotic cells, leading to changes in the surrounding microenvironment that would affect the fluorescence lifetime of Trp-BODIPY. The different localization of Apo-15 in apoptotic vs necrotic cells is likely due to the fact that necrotic cells have compromised membrane permeability and therefore Apo-15 could potentially access different intracellular compartments, whereas the membrane integrity of apoptotic cells limits Apo-15 to binding of PtdSer on the outer leaflet of the plasma membrane.

Line 247-248: It is interesting to see APO-15 distinguishes PS from PE. But please discuss a peptide (RKKWF) has the ability to distinguish them.

Answer: As detailed in our previous answer to reviewer 1, we believe that the stronger binding between Apo-15 and PtdSer may be the result of the favourable electrostatic interactions between the free negatively-charged carboxylate group in PtdSer and the positively-charged residues in Apo-15. This explanation is in agreement with the reduced binding of Apo-15 to PE, which does not have any free negatively-charged carboxylate group. These points are now considered in the revised Discussion of the manuscript (page 11).

Line 269 and Supplementary Figure 13: Figure 13 does not show the DNA fragmentation. The data is very poor.

Answer: We have performed this experiment multiple times and consistently observed DNA degradation but without clear laddering that results from intranucleosomal DNA cleavage.

Therefore, we have used an alternative approach to confirm that the treatment of PLB985 cells with staurosporine induces apoptosis. Using a commercial kit (CellEvent caspase 3/7 probe) for detection of cleaved caspase-3/7, we were able to clearly demonstrate caspase activation in staurosporine-treated PLB985 cells in flow cytometry and confocal microscopy experiments. We have included this new definitive data in the Supplementary Figure 13, replacing the previous DNA gels.

Line 304 and Figure 4A: How the apoptosis was assayed?

Answer: We thank the reviewer for the comment. We have clarified in the text and Figure 4A that the percentage of apoptotic cells was determined by staining with Annexin V.

Line 364-365: Do the authors think that the formalin-treatment does not dissociate APO-15 from PS or it has no effect on APO-15-labelled apoptotic cells that had been engulfed by macrophages?

Answer: We thank the reviewer for the observation. In our experiments we have observed that formalin treatment does not result in a loss of Apo-15-labeling of cells in tissues, and therefore we believe that the interaction between Apo-15 and PtdSer remains stable under the conventional fixation protocols used in tissue processing. On a related note, we have performed a flow cytometry experiment to compare the fluorescence emission of Apo-15 in neutrophils that were fixed with 3% paraformaldehyde (PFA) for 15 min at room temperature after staining as well as non-fixed neutrophils. As shown in the histograms in the Figure for reviewers 2 below, although we observe a minor reduction in mean fluorescence of Apo-15 signal following fixation with paraformaldehyde, Apo-15 staining is broadly similar, indicating that Apo-15 is compatible with conventional fixation protocols.

Figure 2 for reviewers. Flow cytometric analysis of mixtures of apoptotic and viable neutrophils that had been stained with Apo-15 (100 nM) and treated with or without 3% PFA.

Reviewer 3

1. The impact of a novel method lies in scientific questions that were unable to be addressed by previous methods and are now able to be addressed using the novel method. Therefore, to assess the impact of Apo-15, it is critical to know exactly what scientific questions will now be addressable that previously were not. In the context of lung injury, it is not convincing that Apo-15 will allow us to answer new questions. Most studies in fixed lung can already be performed using staining such as cleaved caspase 3 and TUNEL. The authors argue that live imaging is a distinct advantage of the new method but without being able to demonstrate what cell types are Apo-15+ *in vivo* and whether Apo-15 even labels cells other than neutrophils *in vivo*, it is difficult to imagine how this would be useful. This was the concern stated in comment 6 of the initial review. The authors did suggest at least one intriguing example, pulse dosing of Apo-15 to track the fate of apoptotic cells but they acknowledge that more work must be done to show feasibility. The authors argue that additional studies are beyond the scope of this manuscript, and indeed they have done quite a bit of work designing and synthesizing the compound, but without these additional feasibility studies, it is difficult to judge whether Apo-15 will ultimately be useful to answer important scientific questions.

Answer: We thank the reviewer for the comment. We suggest that the extensive experimental characterization we describe in the current manuscript provides a strong foundation for the design of future studies that will test the utility of the Apo-15 probe for detection of apoptosis in different *in vivo* settings. We believe we have discussed our findings in the context of potential limitations and experimental unknowns that are often associated with the development of new technological/methodological approaches. We further suggest that the most robust test of the utility of the Apo-15 would be if other research groups were able to evaluate the probe in a wide variety of *in vitro* or *in vivo* experimental settings.

We have shown that Apo-15 labels apoptotic cells of different origin and under multiple environmental and experimental conditions where other existing reagents do not perform well (e.g. low concentration of calcium ions, live-cell and under physiological conditions where removal of unbound probe would be difficult). We have provided detailed *in vitro* characterisation of Apo-15 with robust evidence that PtdSer represents a major molecular target for Apo-15 binding. Table 1 summarizes these data and highlights some of the key advantages of Apo-15 as an imaging reagent over existing technologies. Out of these, divalent cation independence, functional neutrality and cost effectiveness are features of Apo-15 where Annexin V, the current gold standard for imaging apoptosis, shows clear limitations. As the reviewer will know, it would be difficult to use Annexin V in experiments with cells where calcium concentrations need to be tightly controlled due to potential activation effects (e.g., platelets) or in assays where phagocytosis and cell clearance should remain unaffected (i.e., Annexin V blocks apoptotic cell uptake and drives IFN-dependent anti-tumour responses). The results presented in our manuscript suggest that Apo-15 may be compatible with those 'challenging' experimental conditions. Furthermore, the cost implications of our peptide probe for detection of apoptotic cells may offer considerable benefits. Apo-15 is a small molecule that is relatively cheap to synthesize, therefore facilitating administration protocols *in vivo* that would be prohibitively expensive if using Annexin V.

In our latest revision, we have presented additional evidence that apoptotic epithelial cells can be labelled by Apo-15 and we have shown proof of principle that Apo-15 labels apoptotic cells under real-time imaging. In our opinion, given the evidence that we have presented in this manuscript, it is not a major leap to imagine how this new tool for labeling apoptotic cells *in vivo* (regardless of cell lineage) could be useful for future biological studies. We have

addressed this point in the manuscript text and carefully rewritten the Discussion section (pages 10-13), including examples of possible applications for Apo-15 (e.g., BBB shuttle peptides, tissue regeneration, developmental studies, cardiovascular sciences) and highlighting those that would not be possible with existing reagents.

2. At the bottom of p. 9, the authors state that “BALF and lung digests from LPS-only and LPS+CDKi mice were analyzed by multiparameter flow cytometry” but the digest data is difficult to find.

Answer: We thank the reviewer for the comment. The reviewer is correct as the text should refer to ‘BALF and lung slices’ instead of ‘BALF and lung digests’. We have amended the text in the revised manuscript.

3. In response to concern shared by 2 reviewers that the images were unconvincing, the authors have removed images proposed to show engulfment of apoptotic neutrophils by macrophages. However, the remaining data supporting the claim that Apo-15 labels macrophages that have ingested apoptotic neutrophils is not very strong: In Suppl. Fig. 18, the CD45+ CD11c+ Annexin- cells in the BAL are Apo-15+ when the Apo-15 is administered *in vivo* but these macrophages are not labeled by Apo-15 administered *ex vivo*. The authors argue that these data demonstrate that the macrophages have ingested apoptotic neutrophils. This may be the case but there may be other explanations for these data. Directly demonstrating that these cells have ingested neutrophils by including once again the cytopspins but with staining for macrophage and neutrophil markers would be much stronger.

Answer: We thank the reviewer for the comment. As mentioned in our previous response to the reviewers, we have been unable to co-stain *ex vivo* samples with Apo-15 and antibodies (e.g., Ly6G for neutrophils) given that the processing and antigen retrieval methods that we used in those experiments (e.g., de-paraffinisation and pressure cooking at 105°C in sodium citrate buffer) results in marked loss of the Apo-15 signal, which makes any co-staining with antibody-based markers very difficult.

We believe that our latest data (Supplementary Figures 15 and 18) strongly support our conclusion that Apo-15-labeled macrophages from BALs can arise as a result of efferocytosis of apoptotic cells. Collective, our data shows that:

- 1) BALs of treated mice that were not given Apo-15 show no green fluorescence in viable (i.e., Annexin V-negative) CD45+CD11c+ macrophages, indicating that the staining we observe cannot be due to tissue autofluorescence,
- 2) CD45+CD11c+AnnV- macrophages from BALs of treated mice which had received Apo-15 (5 µM) *ex vivo* show no green fluorescence indicating that Apo-15 does not stain viable macrophages under these conditions, and
- 3) CD45+CD11c+AnnV- macrophages from BALs of treated mice which had received an *in vivo* intratracheal administration of Apo-15 (5 µM) show a strong green fluorescent signal.

Altogether, we believe that the most likely mechanism to explain this staining is the efferocytosis of apoptotic cells which happens *in vivo*. This conclusion is further supported by our *in vitro* observations. First, in Figure 4, we confirmed that Apo-15 does not impair the capacity of human MDMs to engulf dead cells. Secondly, in our last response letter to the reviewers, we showed new data where we observed that green fluorescence staining in phagocytic human MDMs was only detected if Apo-15 was present when phagocytosis occurred (i.e., during incubation with dead cells) but not when phagocytosis had already occurred (i.e., after incubation with dead cells). This experiment eliminates the possibility of

the Apo-15 signal is a result of binding of apoptotic cells or apoptotic cell-derived vesicles to the phagocyte surface. The reviewer suggests that '*this may be the case but there may be other explanations for these data*' but we cannot think of any other possibility that would collectively explain all our results other than macrophages having phagocytosed Apo-15-labeled apoptotic cells or related subcellular material from these cells.

REVIEWERS' COMMENTS:

Reviewer #1 (Remarks to the Author):

I am satisfied by the revision made by the authors.

Reviewer #3 (Remarks to the Author):

The authors have provided reasonable replies to the concerns.

1. The conclusion that macrophages are green by flow because they have ingested green apoptotic cells is based on indirect evidence (Suppl Fig. 18). Direct evidence, similar to that shown in Fig. 4D or cytopins like were included in the initial submission (and would not require antigen retrieval), would be stronger. It is unclear why this didn't work. This limitation could be acknowledged. However, the authors are correct that their interpretation is the most plausible explanation for the data.
2. The authors state that it is not a major leap to imagine how this new tool for labeling apoptotic cells in vivo could be used for future biological studies. In the context of lung injury, since there are so many different cell types in the lung, additional optimization would be necessary to determine in vivo what cell type the Apo-15+ cells are; otherwise, Apo-15 will not be useful. This should be acknowledged.

Response to reviewers (NCOMMS-19-07315C)

Reviewer 3

The authors have provided reasonable replies to the concerns.

1. The conclusion that macrophages are green by flow because they have ingested green apoptotic cells is based on indirect evidence (Suppl Fig. 18). Direct evidence, similar to that shown in Fig. 4D or cytopins like were included in the initial submission (and would not require antigen retrieval), would be stronger. It is unclear why this didn't work. This limitation could be acknowledged. However, the authors are correct that their interpretation is the most plausible explanation for the data.

Answer: We have acknowledged this point in the revised Discussion (page 12) and included the following clarification 'we were unable to co-stain ex vivo samples with **Apo-15** and antibodies given that the processing and antigen retrieval methods results in marked loss of the **Apo-15** signal'.

2. The authors state that it is not a major leap to imagine how this new tool for labeling apoptotic cells in vivo could be used for future biological studies. In the context of lung injury, since there are so many different cell types in the lung, additional optimization would be necessary to determine in vivo what cell type the Apo-15+ cells are; otherwise, Apo-15 will not be useful. This should be acknowledged.

Answer: We have acknowledged this point in the revised Discussion (page 12) and included the following clarification 'further studies would be required to determine the cell types that are preferentially labeled in vivo'.